# The Sparse Matrix-Based Random Projection: A Mean Absolute Deviation Analysis for Sparse Ternary Data

## Abstract

In this paper, we investigate random projections based on sparse $\{0, \pm 1\}$ matrices, which take sparse $\{0, \pm\mu\}$-ternary data as input. Such sparse ternary data, including $\{\pm\mu\}$-binary data as a special case, are widely employed in machine learning for data quantization, and often match or even outperform their full-precision counterparts across various classification tasks. For the projection of such ternary data, we analyze the mean absolute deviation (MAD), a metric that quantifies the dispersion of projected data points. In general, greater dispersion is expected to better capture the intrinsic variations in the original data, making it favorable for downstream classification tasks. Our analysis demonstrates that extremely sparse $\{0, \pm 1\}$ matrices, such as those with only one or a few tens of nonzero entries per row, can achieve large MAD values. By employing such sparse matrices, we indeed obtain favorable classification performance on the projected data. These highly sparse matrix structures suggest that substantial computational savings can be realized in random projection.

## 1 Introduction

In machine learning, data often exhibits large-scale and high-dimensional characteristics, posing considerable challenges to storage and computation. To tackle these issues, two mainstream strategies are widely adopted: quantization and dimensionality reduction. In this paper, we investigate the dimensionality reduction of a specific type of quantized data: sparse $\{0, \pm\mu\}$-ternary data. Such data can be generated by quantizing conventional data features, such as Discrete Cosine Transform (DCT), Discrete Wavelet Transform (DWT), and deep convolutional features (Liu & Liu, 2023). Another prevalent type of quantized data, $\{\pm\mu\}$-binary data, can be regarded as a special case of ternary data. Both types of quantized data have been widely used in various classification tasks without incurring significant performance degradation (Gionis et al., 1999; Ferdowsi et al., 2017). Recent studies have demonstrated that they can achieve comparable or even higher classification accuracy than their full-precision counterparts by enhancing feature discrimination (Lu et al., 2023; 2025).

For these types of quantized data, we investigate dimensionality reduction based on random projection, a method that simply projects high-dimensional data onto low-dimensional subspaces by multiplying data vectors with random matrices (Johnson & Lindenstrauss, 1984). Its low complexity renders it highly suitable for large-scale classification tasks (Gionis et al., 1999; Weinberger et al., 2009; Dalessandro, 2013). While Gaussian matrices are the best known random projection matrices (Dasgupta & Gupta, 1999), our study focus on another promising type: sparse $\{0, \pm 1\}$-ternary matrices (Achlioptas, 2003). Such matrices perform comparably or even better than Gaussian matrices (Lu et al., 2023), while offering significantly lower computational complexity.

The theoretical foundation of random projection for dimensionality reduction lies in its capability to approximately preserve pairwise Euclidean distances between data points with high probability. This helps maintain the global structure of the data, thereby preventing significant performance degradation in downstream tasks such as regression (Nelson & Nguyên, 2013; Clarkson & Woodruff, 2017). For classification tasks, however, strict distance preservation is not the decisive factor; performance depends more on feature discriminability than on exact structural fidelity to the original data. Empirical evidence demonstrates that

strong classification performance can be achieved even when distance preservation is poor. For instance, it is well established that the distance preservation performance of sparse random matrices will deteriorate as the matrices become sparser (i.e. having fewer nonzero entries $\pm 1$) (Li et al., 2006; Matoušek, 2008), particularly in extreme cases such as when each column contains only one nonzero entry (Jagadeesan, 2019). Surprisingly, in classification experiments, these extremely sparse matrices often perform comparably to, and sometimes even better than, their denser counterparts (Lu et al., 2023). This counterintuitive finding challenges conventional interpretations based on distance preservation.

To investigate the impact of random projection on classification, we draw inspiration from principal component analysis (PCA) (Jolliffe, 2002) and focus on analyzing the dispersion of projected data points, namely the variation of projected data points relative to their mean point. A high degree of dispersion indicates that the projected points retain more of the variation inherent in the original data (Jolliffe & Cadima, 2016), making them favorable for downstream classification tasks (Turk & Pentland, 1991). In PCA, dispersion is typically measured using either the $\ell_2$-norm or the $\ell_1$-norm. The $\ell_1$-norm based dispersion measure, known as mean absolute deviation (MAD) (Dodge, 2008), also referred to as L1-PCA (Kwak, 2008) or robust PCA (Deng et al., 2007; McCoy & Tropp, 2011; Meng et al., 2012), often performs better due to its robustness to outliers (Hubert et al., 2016). For this reason, we adopt the MAD metric to quantify the dispersion in random projections.

By varying the number of nonzero entries $\pm 1$ in the sparse random projection matrices, we estimate the MAD value of the projected data and observe two intriguing trends. Firstly, when the original ternary data are sufficiently dense, meaning they contain relatively few zero entries, the MAD reaches its maximum when each row of the sparse random matrix contains only one nonzero entry. Secondly, the MAD value tends to rapidly converge to a stable level as the number of nonzero entries $\pm 1$ in the random projection matrix increases. Collectively, these two trends suggest that extremely sparse random projection matrices, such as those with only a single or a few dozen non-zero entries per row, can achieve classification performance comparable to or even superior to that of denser matrices. This brings substantial computational savings for random projections. This property is validated through classification experiments conducted on ternary data generated by quantizing common features of real-world datasets. Furthermore, it is noteworthy that the MAD trends and related classification performance described above hold not only for ternary quantized data but also in the presence of Gaussian noise. This underscores the robustness and generalizability of our theoretical findings.

## 2 Problem Formulation

Given a random projection $\boldsymbol{z} = \boldsymbol{R}\boldsymbol{x}$, where $\boldsymbol{x} \in \{0, \pm\mu\}^n$ denotes a sparse ternary data and $\boldsymbol{R} \in \{0, \pm 1\}^{m \times n}$ is a sparse random matrix. Our goal is to estimate the mean absolute deviation (MAD) of the projection $\boldsymbol{z} \in \mathbb{R}^m$, which is formally expressed as $\text{MAD}(\boldsymbol{z}) = \mathbb{E}\|\boldsymbol{z} - \mathbb{E}\boldsymbol{z}\|_1$. In the following, we first specify the distributions of the sparse matrix $\boldsymbol{R}$ and the sparse data $\boldsymbol{x}$, and then elaborate the form of MAD.

**Notation.** Throughout the paper, we typically denote a matrix by a bold upper-case letter $\boldsymbol{R} \in \mathbb{R}^{m \times n}$, a vector by a bold lower-case letter $\boldsymbol{r} = (r_1, r_2, ..., r_n)^\top \in \mathbb{R}^n$, and a scalar by a lower-case letter $r_i$ or $r$. In some cases, we use the bold letter $\boldsymbol{r}_i \in \mathbb{R}^n$ to denote the $i$-th row of $\boldsymbol{R} \in \mathbb{R}^{m \times n}$. The expectation function is written as $\mathbb{E}(\cdot)$. For ease of presentation, we defer all proofs to the appendix.

### 2.1 Sparse random matrices

The sparse random matrix $\boldsymbol{R}$ is specified in Definition 1, which has the parameter $k$ counting the number of nonzero entries per row, and is simply called $k$-sparse to distinguish between the matrices of different sparsity. Instead of the form $\boldsymbol{R} \in \{0, \pm 1\}^{m \times n}$, in the definition we introduce a scaling parameter $\sqrt{\frac{n}{mk}}$ to make the matrix entries have zero mean and unit variance. With such distribution, the matrix can satisfy the desired distance preservation property, that is, having $\mathbb{E}\|\boldsymbol{R}\boldsymbol{x}\|_2 = \mathbb{E}\|\boldsymbol{x}\|_2$ with relatively high probability (Achlioptas, 2003). For easier computation, the scaling parameter can be omitted in practical applications, while the omitting will not change the relative distances between projected data points, thus not affecting downstream classification tasks.

**Definition 1** (*k*-sparse random matrix). A *k*-sparse random matrix $\boldsymbol{R} \in \{0, \pm\sqrt{\frac{n}{mk}}\}^{m \times n}$ is defined to be of the following structure properties:

- Each row $\boldsymbol{r} \in \{0, \pm\sqrt{\frac{n}{mk}}\}^n$ contains exactly $k$ nonzero entries, $1 \le k \le n$;

- The positions of $k$ nonzero entries are arranged uniformly at random;

- Each nonzero entry takes the bipolar values $\pm\sqrt{\frac{n}{mk}}$ with equal probability.

### 2.2 Sparse ternary data

**Data distribution.** For the ternary data $\boldsymbol{x} = (x_1, x_2, \ldots, x_n)^\top$, we assume its each entry $x_i$ independently follows the distribution

$$x_i \sim \mathcal{T}(\mu, p, q) \tag{1}$$

which has the probability mass function $t \in \{-\mu, 0, \mu\}$ under the probabilities $\{q, p, q\}$, where $\mu$ is a positive constant and $p + 2q = 1$. It is seen that a larger $p$ value (equivalently, a smaller $q$ value) indicates a greater number of zero entries in $x_i$, suggesting a sparser distribution.

**Data generation.** As previously mentioned, in practical scenarios, such ternary data can be generated by quantizing data features using an element-wise thresholding approach, and usually can achieve comparable or even better classification performance than the original features (Lu et al., 2023; 2025).

**Gaussian noise.** When further adding zero-mean Gaussian noise to the ternary data $\boldsymbol{x}$ defined in (1), its each entry $x_i$ will follow a three-component Gaussian mixture distribution

$$x_i \sim \mathcal{M}(\mu, \sigma^2, p, q) \tag{2}$$

with the probability density function

$$f(t) = pf_\mathcal{N}(t; 0, \sigma^2) + qf_\mathcal{N}(t; \mu, \sigma^2) + qf_\mathcal{N}(t; -\mu, \sigma^2) \tag{3}$$

where $f_\mathcal{N}(t; \mu, \sigma^2)$ denotes the density function of the variable $t \sim \mathcal{N}(\mu, \sigma^2)$, and $p$ and $q$ represent the mixture weights of three components, provided $p + 2q = 1$ and $p, q \ge 0$. Similarly as in the noiseless case (1), a larger $p$ value indicates a sparser data structure. In practice, the original data features used for ternary quantization can be roughly regarded as following such a Gaussian mixture distribution, since quantization errors are typically modeled as additive Gaussian noise (Gray & Neuhoff, 2002). Therefore, in the final experiments we also examine the classification performance for the projections of the original full-precision data features.

### 2.3 Mean absolute deviation

For a random projection $\boldsymbol{z} = \boldsymbol{R}\boldsymbol{x}$, its mean absolute deviation (MAD) is defined as $\mathrm{MAD}(\boldsymbol{z}) = \mathbb{E}\|\boldsymbol{z} - \mathbb{E}\boldsymbol{z}\|_1$. As detailed later, the MAD can be further formulated as

$$\mathrm{MAD}(\boldsymbol{z}) = \mathbb{E}\|\boldsymbol{R}\boldsymbol{x}\|_1 = m\mathbb{E}|\boldsymbol{r}^\top\boldsymbol{x}| \tag{4}$$

where $\boldsymbol{r} \in \mathbb{R}^n$ denotes a row of $\boldsymbol{R} \in \{0, \pm\sqrt{\frac{n}{mk}}\}^{m \times n}$ and $m$ is the projection dimension of $\boldsymbol{R}$. When the size $m \times n$ of $\boldsymbol{R}$ is given, a larger $\mathbb{E}|\boldsymbol{r}^\top\boldsymbol{x}|$ will correspond to a higher $\mathrm{MAD}(\boldsymbol{z})$ value. By this equivalence relationship, our estimation of $\mathrm{MAD}(\boldsymbol{z})$ is simplified to an estimation focused on $\mathbb{E}|\boldsymbol{r}^\top\boldsymbol{x}|$ in the subsequent theoretical analysis.

Below, we detail the derivation of (4). Consider a random projection $\boldsymbol{z} = \boldsymbol{R}\boldsymbol{x}$, where $\boldsymbol{R}$ follows the distribution specified in the Definition 1 and $\boldsymbol{x}$ is distributed as given in either (1) or (2). The MAD of $\boldsymbol{z}$ can be written as $\mathrm{MAD}(\boldsymbol{z}) = \mathbb{E}\|\boldsymbol{R}\boldsymbol{x} - \mathbb{E}(\boldsymbol{R}\boldsymbol{x})\|_1$. Since the entries of $\boldsymbol{R}$ and $\boldsymbol{x}$ are independent and symmetric about zero, it straightforwardly follows that $\mathbb{E}(\boldsymbol{R}\boldsymbol{x}) = 0$. Consequently, the MAD of $\boldsymbol{z}$ is reduced to $\mathrm{MAD}(\boldsymbol{z}) = \mathbb{E}\|\boldsymbol{R}\boldsymbol{x}\|_1$. Furthermore, because each row $\boldsymbol{r}$ of $\boldsymbol{R}$ is independently and identically distributed, we can express $\mathbb{E}\|\boldsymbol{R}\boldsymbol{x}\|_1 = m\mathbb{E}|\boldsymbol{r}^\top\boldsymbol{x}|$, where $m$ denotes the number of rows in $\boldsymbol{R}$. Then (4) is derived.

## 3 Theoretical Results

### 3.1 Sparse ternary data

For the random projection $\boldsymbol{z} = \boldsymbol{R}\boldsymbol{x}$ with $\boldsymbol{x}$ being sparse ternary data distributed as in (1), Theorem 1 estimates its mean absolute deviation MAD($\boldsymbol{z}$) by using the equivalent form $\mathbb{E}|\boldsymbol{r}^\top \boldsymbol{x}|$. As described in (4), given the size $m \times n$ of random matrix $\boldsymbol{R}$, a larger value of $\mathbb{E}|\boldsymbol{r}^\top \boldsymbol{x}|$ corresponds to a larger MAD($\boldsymbol{z}$).

**Theorem 1.** Let $\boldsymbol{r}$ be a row of a $k$-sparse random matrix $\boldsymbol{R} \in \{0, \pm\sqrt{\frac{n}{mk}}\}^{m \times n}$ as specified in Definition 1, and $\boldsymbol{x} \in \mathbb{R}^n$ with i.i.d. entries $x_i \sim \mathcal{T}(\mu, p, q)$ as detailed in (1). It is derived that

$$\mathbb{E}|\boldsymbol{r}^\top \boldsymbol{x}| = 2\mu\sqrt{\frac{n}{mk}} \sum_{i=0}^{k} C_k^i p^i q^{k-i} \left\lceil \frac{k-i}{2} \right\rceil C_{k-i}^{\lceil \frac{k-i}{2} \rceil} \tag{5}$$

and

$$\text{Var}(|\boldsymbol{r}^\top \boldsymbol{x}|) = \frac{2q\mu^2 n}{m} - \frac{4\mu^2 n}{mk} \left( \sum_{i=0}^{k} C_k^i p^i q^{k-i} \left\lceil \frac{k-i}{2} \right\rceil C_{k-i}^{\lceil \frac{k-i}{2} \rceil} \right)^2 \tag{6}$$

where $C_k^i$ is a binomial coefficient $\binom{k}{i}$ and $\lceil \alpha \rceil = \min\{\beta : \beta \geq \alpha, \beta \in \mathbb{Z}\}$. By (5), $\mathbb{E}|\boldsymbol{r}^\top \boldsymbol{x}|$ satisfies the following two properties:

(P1) When $p \leq 0.188$, $\mathbb{E}|\boldsymbol{r}^\top \boldsymbol{x}|$ has its maximum at $k = 1$.

(P2) When $k \to \infty$, $\mathbb{E}|\boldsymbol{r}^\top \boldsymbol{x}|$ converges to a constant:

$$\lim_{k \to \infty} \frac{\sqrt{m}}{\mu\sqrt{n}} \mathbb{E}|\boldsymbol{r}^\top \boldsymbol{x}| = 2\sqrt{q/\pi}, \tag{7}$$

which has the convergence error upper-bounded by

$$\left| \frac{\sqrt{m}}{\mu\sqrt{n}} \mathbb{E}|\boldsymbol{r}^\top \boldsymbol{x}| - 2\sqrt{q/\pi} \right| \leq \frac{\sqrt{\pi} + \sqrt{2}}{\sqrt{\pi k}}, \tag{8}$$

for finite $k$ values.

The results provided in P1 and P2 demonstrate two interesting trends of $\mathbb{E}|\boldsymbol{r}^\top \boldsymbol{x}|$ against the varying sparsity $k$ of random matrix $\boldsymbol{R}$, where $k$ counts the number of nonzero entries in each row of $\boldsymbol{R}$. These trends are further discussed as follows.

**Regarding P1:** This finding indicates that $\mathbb{E}|\boldsymbol{r}^\top \boldsymbol{x}|$ reaches its maximum value when $k = 1$, provided the probability $p$ of each entry $x_i$ in $\boldsymbol{x}$ being zero value is sufficiently low, specifically $p \leq 0.188$. In other words, the ternary data $\boldsymbol{x}$ needs to exhibit a sufficiently dense distribution. Given that a larger $\mathbb{E}|\boldsymbol{r}^\top \boldsymbol{x}|$ (equivalent to a larger MAD($\boldsymbol{z}$) probably yields better classification performance, it is expected that the best classification performance for the projected data can be achieved using extremely sparse random matrices with sparsity $k = 1$ (meaning each row contains only one nonzero entry), when the ternary data $\boldsymbol{x}$ is sufficiently dense.

**Regarding P2:** The result in (7) reveals that given the distribution of $\boldsymbol{x}$ and the size of $\boldsymbol{R}$, $\mathbb{E}|\boldsymbol{r}^\top \boldsymbol{x}|$ will converge to a constant value as the sparsity $k$ of $\boldsymbol{R}$ approaches infinity. As will be analyzed later, the convergence rate is rapid, and small convergence errors can be attained even when $k$ is relatively small, on the order of tens. This suggests that sparse ternary matrices with only a few tens of nonzero entries per row can achieve classification performance comparable to that of their denser counterparts. Below, we elaborate on the convergence property.

In (8), we investigate the error in the convergence of $\mathbb{E}|\boldsymbol{r}^\top \boldsymbol{x}|$ to the constant value $2\sqrt{q/\pi}$, as derived in (7). Here, the scaling parameter $\sqrt{m}/(\mu\sqrt{n})$ can be treated as a constant and ignored, given the nonzero value $\mu$ of $\mathbf{x}$ and the size $m \times n$ of $R$. It is seen that the convergence error decreases at a rate of $\mathcal{O}(\sqrt{k})$.

This implies that as the matrix sparsity $k$ increases, the convergence error decreases. Furthermore, for any specified error rate $\eta$, we can establish a lower bound for $k$ as follows

$$\frac{\left|\frac{\sqrt{m}}{\mu\sqrt{n}}\mathbb{E}|\boldsymbol{r}^\top\boldsymbol{x}| - 2\sqrt{q/\pi}\right|}{2\sqrt{q/\pi}} \leq \eta, \quad \text{if}\ \ k \geq \frac{(\sqrt{\pi} + \sqrt{2})^2}{4q\eta^2} \tag{9}$$

which is obtained by transforming (8). This lower bound of $k$ can be explicitly determined and decreases with increasing $q$ (corresponding to denser data distributions). Specifically, from (9), the lower bound of $k$ is derived as 5078, 1016 and 565 for $q \in \{0.05, 0.25, 0.45\}$ (or equivalently $p \in \{0.9, 0.5, 0.1\}$), provided $\mu = 1$ and $\eta = 0.1$. Note that this theoretical bound is rather loose; the actual value is much lower, typically on the order of tens, as validated by the numerical results in Figure 6(a), Appendix B.1.

**Numerical validation.** To further verify the correctness of our theoretical findings P1 and P2, in Appendix B.1 we investigate how the $\mathbb{E}|\mathbf{r}^\top\mathbf{x}|$ value varies with matrix sparsity $k$ through two methods: 1) by directly computing $\mathbb{E}|\mathbf{r}^\top\mathbf{x}|$ using (10), and 2) by statistically estimating $\mathbb{E}|\mathbf{r}^\top\mathbf{x}|$ through synthetic data. Both methods produce results that align with P1 and P2.

Moreover, it is noteworthy that the $\mathrm{MAD}(\boldsymbol{z})$ analyzed above represents an expected value $\mathbb{E}\|\boldsymbol{R}\boldsymbol{x}\|_1$ (equivalently, $m\mathbb{E}|\boldsymbol{r}^\top\boldsymbol{x}|$), rather than the actual value $\|\boldsymbol{R}\boldsymbol{x}\|_1$ that could be obtained with a specific matrix. To approximate this expected value and then achieve the theoretical performance demonstrated in P1 and P2, a single matrix should have the size of $m \geq \mathcal{O}(\sqrt{n})$, as proved in Property 1.

**Property 1.** Let $\boldsymbol{r}_i$ be the $i$-th row of a $k$-sparse random matrix $\boldsymbol{R} \in \{0, \pm\sqrt{\frac{n}{mk}}\}^{m \times n}$, and $\boldsymbol{x} \in \mathbb{R}^n$ with i.i.d. entries $x_i \sim \mathcal{T}(\mu, p, q)$. Suppose $z = \frac{1}{m}\|\boldsymbol{R}\boldsymbol{x}\|_1 = \frac{1}{m}\sum_{i=1}^{m}|\boldsymbol{r}_i^\top\boldsymbol{x}|$. For arbitrarily small $\varepsilon, \delta > 0$, we have the probability $\Pr\{|z - \mathbb{E}z| \leq \varepsilon\} \geq 1 - \delta$, if $\frac{m^2}{m+1} \geq \frac{q\mu^2 n}{\varepsilon^2\delta}$; and the condition can be relaxed to $m^2 \geq \frac{2q\mu^2 n}{\varepsilon^2\delta}$, for a given $\boldsymbol{x}$.

## 3.2 Sparse ternary data with Gaussian noise

In Theorem 2, we investigate the $\mathrm{MAD}(\boldsymbol{z})$ of $\boldsymbol{z} = \boldsymbol{R}\boldsymbol{x}$ by leveraging its equivalent form $\mathbb{E}|\boldsymbol{r}^\top\boldsymbol{x}|$, where $\boldsymbol{x}$ denotes sparse ternary data with Gaussian noise added as specified in (2).

**Theorem 2.** Let $\boldsymbol{r}$ be a row of a $k$-sparse random matrix $\boldsymbol{R} \in \{0, \pm\sqrt{\frac{n}{mk}}\}^{m \times n}$ as specified in Definition 1, and $\boldsymbol{x} \in \mathbb{R}^n$ with i.i.d. entries $x_i \sim \mathcal{M}(\mu, \sigma^2, p, q)$ as detailed in (2). It is derived that

$$\mathbb{E}|\boldsymbol{r}^\top\boldsymbol{x}| = 2\mu\sqrt{\frac{n}{mk}}T_1 + \sigma\sqrt{\frac{2n}{\pi m}}T_2 - 2\mu\sqrt{\frac{n}{mk}}T_3 \tag{10}$$

$$T_1 = \sum_{i=0}^{k} C_k^i p^i q^{k-i} \left\lceil \frac{k-i}{2} \right\rceil C_{k-i}^{\lceil\frac{k-i}{2}\rceil}$$

$$T_2 = \sum_{i=0}^{k} C_k^i p^i q^{k-i} \sum_{j=0}^{k-i} C_{k-i}^j e^{-\frac{(k-i-2j)^2\mu^2}{2k\sigma^2}}$$

$$T_3 = \sum_{i=0}^{k} C_k^i p^i q^{k-i} \sum_{j=0}^{k-i} C_{k-i}^j \Phi\left(-\frac{|k-i-2j|\mu}{\sqrt{k}\sigma}\right)$$

and

$$\mathrm{Var}(|\boldsymbol{r}^\top\boldsymbol{x}|) = \frac{n}{m}(\sigma^2 + 2q\mu^2) - \left(\mathbb{E}|\boldsymbol{r}^\top\boldsymbol{x}|_1\right)^2 \tag{11}$$

where $\Phi(\cdot)$ is the distribution function of $\mathcal{N}(0,1)$. Further, we have

$$\mathbb{E}|\boldsymbol{r}^\top\boldsymbol{x}| \leq \mu\sqrt{\frac{n}{m}} + \sigma\sqrt{\frac{2n}{\pi m}}, \tag{12}$$

and

$$\lim_{k\to\infty} \frac{\sqrt{m}}{\mu\sqrt{n}}\mathbb{E}|\boldsymbol{r}^\top\boldsymbol{x}| = \sqrt{\frac{2}{\pi}(\sigma^2 + 2q\mu^2)} \tag{13}$$

which has the convergence error for finite $k$ upper-bounded by

$$\left| \frac{\sqrt{m}}{\mu\sqrt{n}} \mathbb{E}|\boldsymbol{r}^\top \boldsymbol{x}| - \sqrt{2(\sigma^2 + 2q\mu^2)/\pi} \right| \leq \frac{4\sigma^3 \left[p + 2q(1 + \mu^2/\sigma^2)^{3/2}\right]}{(\sigma^2 + 2q\mu^2)\sqrt{\pi k}} + \frac{\sqrt{2}[3\sigma^4 + 2q(6\sigma^2\mu^2 + \mu^4)]}{\sqrt{(\sigma^2 + 2q\mu^2)\pi k}}. \quad (14)$$

The results demonstrate that $\mathbb{E}|\boldsymbol{r}^\top \boldsymbol{x}|$ exhibits trends similar to those described in P1 and P2 of Theorem 1, as the sparsity $k$ of random matrix $\boldsymbol{R}$ varies. These similarities are discussed below.

**Similar to P1:** By directly computing (10), as detailed in Appendix B.2, we observe that $\mathbb{E}|\boldsymbol{r}^\top \boldsymbol{x}|$ attains its maximum value at $k = 1$, when the parameters $p$ and $\sigma$ of $\boldsymbol{x}$ (as specified in (2)) are relatively small, specifically, when $p < 1/2$ and $\sigma/\mu < 1/3$. A smaller $p$ corresponds to a denser data distribution, while a smaller $\sigma/\mu$ indicates a relatively low level of Gaussian noise.

**Similar to P2:** From (13) and (14), it can be seen that $\mathbb{E}|\boldsymbol{r}^\top \boldsymbol{x}|$ converges to a constant as the matrix sparsity $k$ increases, with a convergence rate of $\mathcal{O}(\sqrt{k})$. Using (14), we can readily derive a lower bound of $k$ that ensures the convergence error rate below a given constant $\eta$:

$$\frac{\left| \frac{\sqrt{m}}{\mu\sqrt{n}} \mathbb{E}|\boldsymbol{r}^\top \boldsymbol{x}| - \sqrt{2(\sigma^2 + 2q\mu^2)/\pi} \right|}{\sqrt{2(\sigma^2 + 2q\mu^2)/\pi}} \leq \eta, \quad (15)$$

provided $k \geq \left( \frac{4\sigma^3[p + 2q(1 + \mu^2/\sigma^2)^{3/2}]}{(\sigma^2 + 2q\mu^2)^{3/2}\sqrt{2}\eta} + \frac{3\sigma^4 + 2q(6\sigma^2\mu^2 + \mu^4)}{(\sigma^2 + 2q\mu^2)\eta} \right)^2$. As the standard deviation $\sigma$ of Gaussian noise converges to zero, the above lower bound for $k$ will approach the bound established for noiseless ternary data in (9). Furthermore, a relatively modest error rate $\eta$ can be achieved with a small value of $k$, such as tens, as illustrated in Figure 6(b), Appendix B.1.

Overall, these properties indicate that sparse ternary data will preserve its performance trends in terms of MAD and classification, provided that the level of additive Gaussian noise is not excessively high. As previously noted, this noise model can roughly approximate the distribution of the original data from which ternary data are generated. Consequently, our theoretical insights about ternary data are also partially applicable to real-world data, as verified in the final experiments.

As was done for Theorem 1, we validate the correctness of Theorem 2 by conducting additional numerical computations and statistical simulations in Appendix B.2. Moreover, to ensure the MAD value $\mathbb{E}\|\boldsymbol{R}^\top \boldsymbol{x}\|_1$ close to the actual observation $\|\boldsymbol{R}^\top \boldsymbol{x}\|_1$ from a specific matrix, the matrix should satisfy the size of $m \geq \mathcal{O}(\sqrt{n})$. The analysis is similar to that in Property 1, omitted here.

## 4 Experiments

In this section, we aim to verify that extremely sparse $\{0, \pm 1\}$-ternary random matrices, which contain only one or a few nonzero entries per row, can achieve classification performance comparable, or even superior to that of denser ternary matrices, as predicted by the MAD analyses provided in Theorems 1 and 2.

### 4.1 Setting

For the sake of generality, we evaluate classification performance using diverse data features across datasets of varying sacles. These include DCT features extracted from YaleB (Georghiades et al., 2001; Lee et al., 2005), deep convolutional features from ResNet18 (He et al., 2016) on CIFAR10 (Krizhevsky & Hinton, 2009), and vision-transformer (ViT-B/32) features (Dosovitskiy et al., 2021) on CIFAR100 and ImageNet1000 (Deng et al., 2009). Classification is conducted both on the original full-precision features and their ternary quantization counterparts, which serve to validate Theorems 2 and 1, respectively. Following the approach in (Lu et al., 2023; 2025), ternary features are generated by quantizing each element of the *standardized* feature vectors to the values $\{-1, 0, 1\}$ using thresholds $\{-\tau, \tau\}$. Here, $\tau$ is calculated as the mean of the absolute values of all elements in each feature vector.

To further ensure generality, we conduct both binary classification on relatively small datasets YaleB and CIFAR10 via exhaustive enumeration of all class pairs, and more challenging multi-class classification on

the more complex datasets CIFAR100 and ImageNet1000. To directly reflect the discriminability of the projected data, we adopt the naive $K$-nearest neighbor ($K$NN) classifier (Cover & Hart, 1967), which relies solely on pairwise data similarities without additional discrimination enhancement. To investigate the impact of random projection matrix distribution on classification performance, we vary the matrix sparsity level $k$, namely the number of nonzero entries per row, within the range $k \in [1, 30]$ under three distinct compression ratios $m/n \in \{1\%, 10\%, 50\%\}$.

## 4.2 Results

We provide the binary classification results for YaleB and CIFAR10 in Figures 1 and 2, respectively, and the multiclass classification results for CIFAR100 and ImageNet1000 in Figures 3 and 4. Overall, the results are consistent with the MAD analyses given in Theorems 1 and 2. The consistency mainly manifests in the following several aspects.

Firstly, as predicted by the theoretical results in (7) and (13), the classification performance indeed converges rapidly to a stable level as the sparsity $k$ of random projection matrices increases. Specifically, convergence can usually be achieved at a small value of $k < 10$, which aligns with the numerical analyses presented in Figure 6. This indicates that extremely sparse matrices can achieve performance comparable to that of denser matrices, a promising finding for the efficient implementation of sparse random matrices in practical scenarios.

Secondly, a comparison between the results of original and ternary features reveals that full-precision data tend to reach convergence at smaller values of $k$. By the analysis of (9), this phenomenon arises because data with larger values of parameter $q$, i.e., those with denser distributions, enable faster convergence. Evidently, the distribution of original full-precision data is denser than that of their ternary quantized counterparts.

Thirdly, as the compression ratio $m/n$ increases, convergence tends to be achieved at smaller values of $k$, a trend that better aligns with our theoretical prediction. As analyzed in Property 1, this is because an increasing compression ratio $m/n$ enables the actual random projection matrices to better approximate the theoretically estimated MAD values.

Finally, when the compression ratio is sufficiently large, such as $m/n = 0.5$, classification on the original full-precision data typically achieve better performance at $k = 1$ than at larger values of $k$. These results are consistent with the analysis of Theorem 2. As indicated by this analysis, the maximum MAD value can be attained at $k = 1$ when the data distribution has a sufficiently small parameter $p$, i.e., when the data is sufficiently dense. Such dense distributions should be exhibited by the original full-precision data.

In addition, as observed in (Lu et al., 2023), our classification results demonstrate that extremely sparse random matrices can achieve performance comparable to Gaussian random matrices, while incurring substantially lower computational complexity. This makes them a viable alternative to Gaussian matrices in practical applications.

## 5 Conclusion

For random projections based on sparse ternary matrices, we present the first analysis of the mean absolute deviation (MAD). The analysis reveals that extremely sparse ternary matrices, which contain only one or a small number of nonzero entries per row, can achieve MAD values comparable to, or even superior to, those of denser ternary matrices. This holds true when the data being projected follows a ternary discrete distribution, even in the presence of Gaussian noise. The MAD property indicates that such matrices can deliver robust classification performance despite their high sparsity. This capability is validated through both binary and multi-class classification experiments conducted on commonly used data features, including DCT features from YaleB, deep convolutional features from CIFAR10, transformer-derived features from CIFAR100 and ImageNet1000, as well as their ternary-quantized variants. Given the extensive use of random projection in dimensionality reduction (Charikar, 2002) and its critical role in deep neural network modeling (Giryes et al., 2016; Rokh et al., 2023), our findings provide valuable insights for reducing computational and model complexity in related applications.

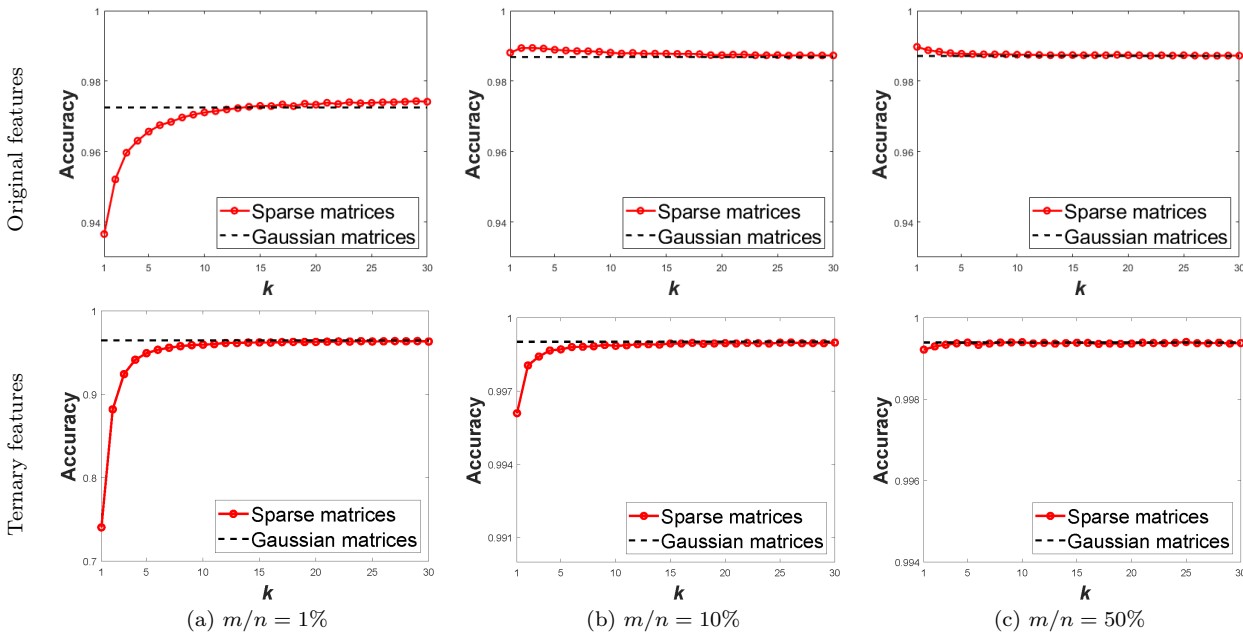

Figure 1: Binary classification accuracy of random projections for original and ternary features from YaleB (DCT features). The sparse random projection matrix has its sparsity level varied across the range $k \in [1, 30]$ under three distinct compression ratios $m/n \in \{1\%, 10\%, 50\%\}$. For reference, the performance of random projections based on Gaussian matrices is included as a baseline.

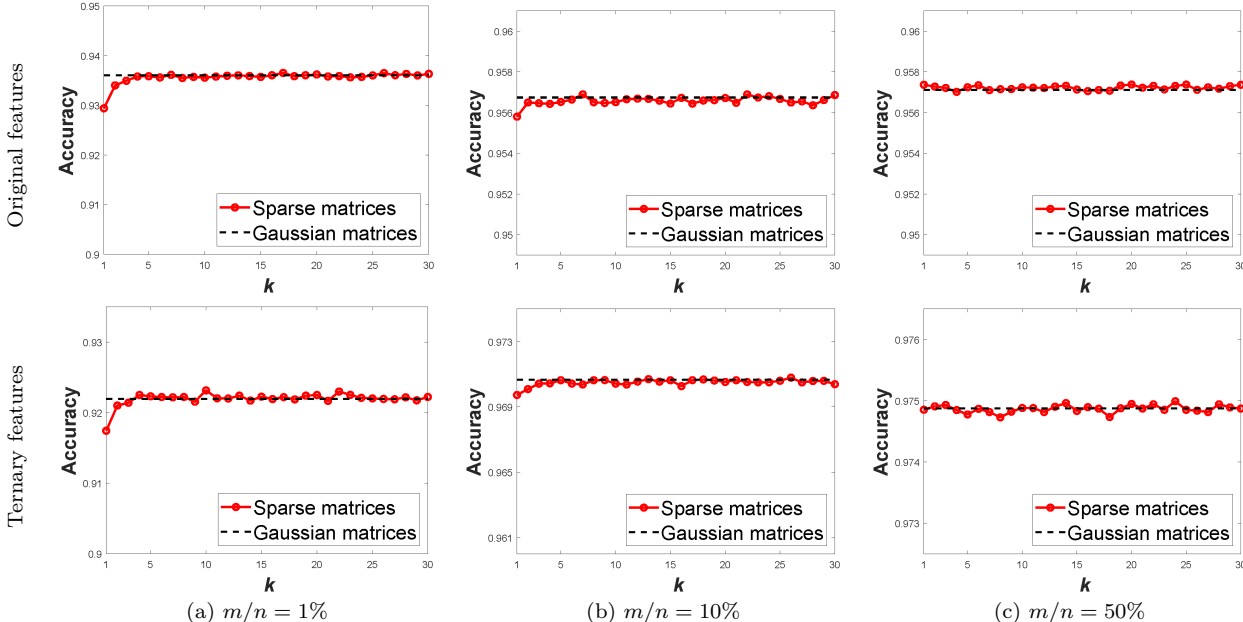

Figure 2: Binary classification accuracy of random projections for original and ternary features from CIFAR10 (ResNet18 features). The sparse random projection matrix has its sparsity level varied across the range $k \in [1, 30]$ under three distinct compression ratios $m/n \in \{1\%, 10\%, 50\%\}$. For reference, the performance of random projections based on Gaussian matrices is included as a baseline.

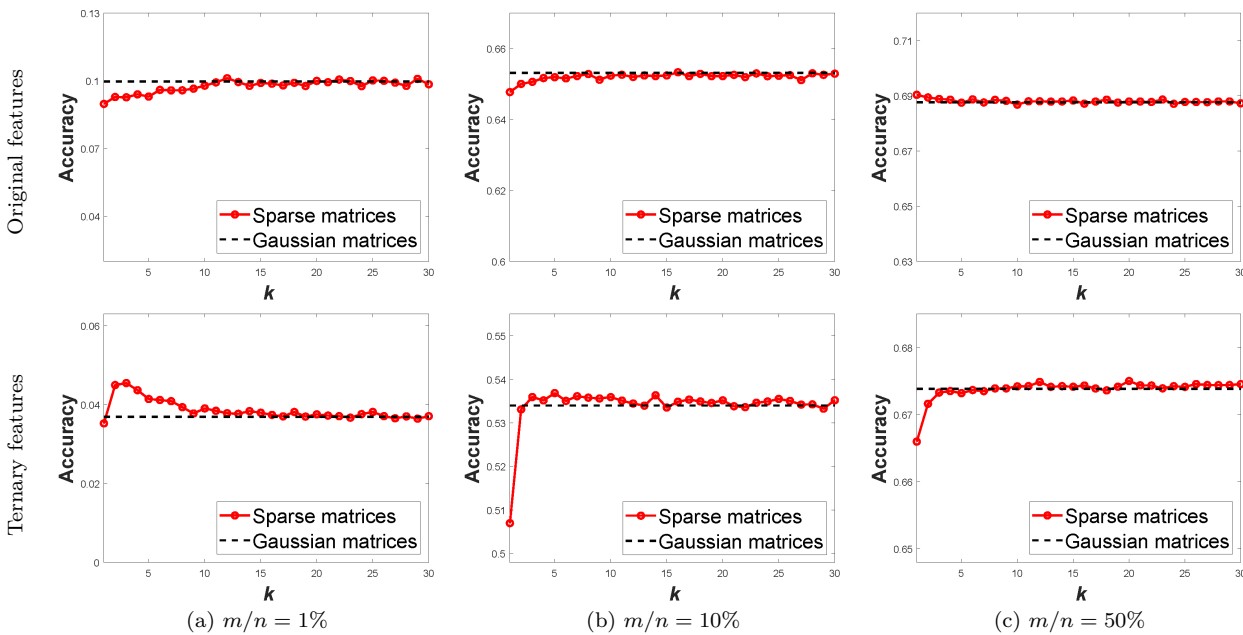

Figure 3: Multiclass classification accuracy of random projections for original and ternary features from CIFAR100 (ViT-B/32 features). The sparse random projection matrix has its sparsity level varied across the range $k \in [1, 30]$ under three distinct compression ratios $m/n \in \{1\%, 10\%, 50\%\}$. For reference, the performance of random projections based on Gaussian matrices is included as a baseline.

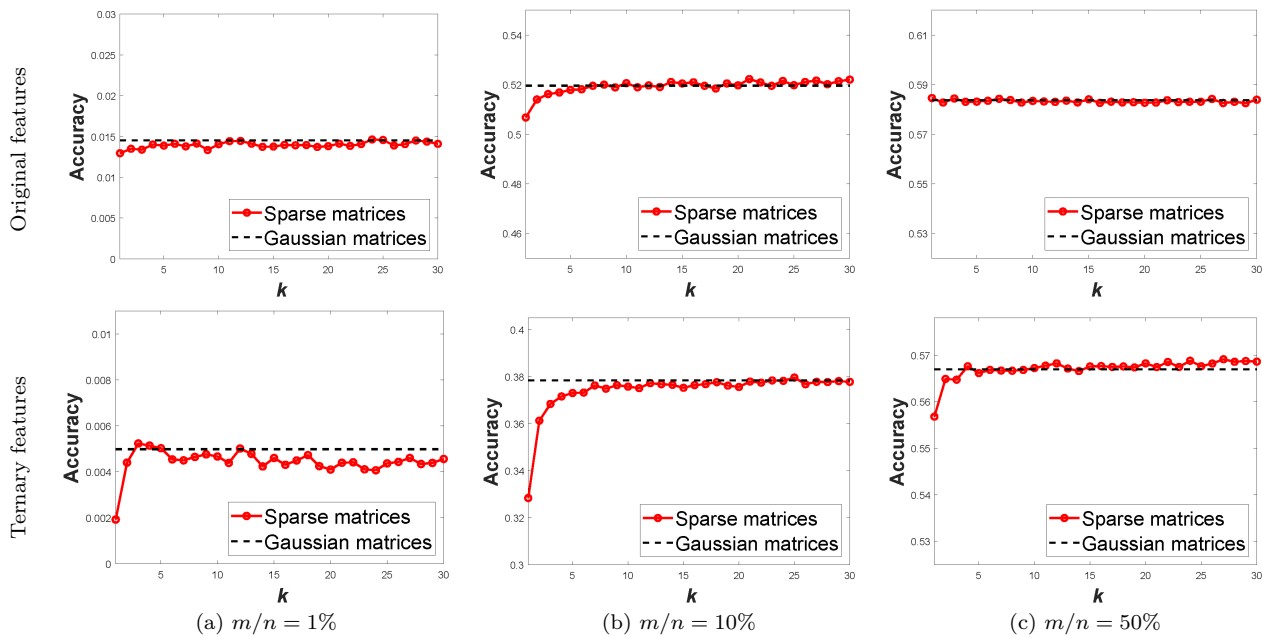

Figure 4: Multiclass classification accuracy of random projections for original and ternary features from ImageNet1000 (ViT-B/32 features). The sparse random projection matrix has its sparsity level varied across the range $k \in [1, 30]$ under three distinct compression ratios $m/n \in \{1\%, 10\%, 50\%\}$. For reference, the performance of random projections based on Gaussian matrices is included as a baseline.

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

# A Proofs for Theorems 1-2 and Property 1 in Section 3

## A.1 Proof of Theorem 1

*Proof.* In the following, we sequentially prove (5), (6), P1 and P2.

**Proofs of** (5) **and** (6)**:** With the distributions of $\boldsymbol{r}$ and $\boldsymbol{x}$, we can write $\|\boldsymbol{r}^\top \boldsymbol{x}\|_1 = \sqrt{\frac{n}{mk}}\mu \left|\sum_{i=1}^k z_i\right|$, where $z_i \in \{-1, 0, 1\}$ with probabilities $\{q, p, q\}$. Then, it can be derived that

$$\mathbb{E}|\boldsymbol{r}^\top \boldsymbol{x}| = \mu\sqrt{\frac{n}{mk}}\sum_{i=0}^k C_k^i p^i q^{k-i}\sum_{j=0}^{k-i} C_{k-i}^j |k - i - 2j|, \tag{16}$$

among which $\sum_{j=0}^{k-i} C_{k-i}^j |k - i - 2j|$ can be expressed as

$$\sum_{j=0}^{k-i}(C_{k-i}^j |k - i - 2j|) = 2\left\lceil\frac{k-i}{2}\right\rceil C_{k-i}^{\lceil\frac{k-i}{2}\rceil}, \tag{17}$$

where $\lceil\alpha\rceil = \min\{\beta : \beta \geq \alpha, \beta \in \mathbb{Z}\}$. Combine (16) and (17), we can obtain (5).

Next, we can derive the variance of $|\boldsymbol{r}^\top \boldsymbol{x}|$

$$\begin{aligned}\mathrm{Var}(|\boldsymbol{r}^\top \boldsymbol{x}|) &= \mathrm{Var}(\boldsymbol{r}^\top \boldsymbol{x}) - \left(\mathbb{E}|\boldsymbol{r}^\top \boldsymbol{x}|\right)^2 \\ &= \frac{2q\mu^2 n}{m} - \frac{4\mu^2 n}{mk}\left(\sum_{i=0}^k C_k^i p^i q^{k-i}\left\lceil\frac{k-i}{2}\right\rceil C_{k-i}^{\lceil\frac{k-i}{2}\rceil}\right)^2.\end{aligned} \tag{18}$$

**Proof of P1:** This part aims to prove

$$\mathbb{E}|\boldsymbol{r}^\top \boldsymbol{x}|_{k=1} > \mathbb{E}|\boldsymbol{r}^\top \boldsymbol{x}|_{k>1},$$

where the subscript $k = 1$ denotes the case of $\mathbb{E}|\boldsymbol{r}^\top \boldsymbol{x}|$ with $k = 1$, and the subscript $k > 1$ means the case of $k$ taking any integer value greater than 1. In the following, we will calculate and compare $\mathbb{E}|\boldsymbol{r}^\top \boldsymbol{x}|$ in terms of the two cases. For the case of $k = 1$, by (5), it is easy to derive that

$$\mathbb{E}|\boldsymbol{r}^\top \boldsymbol{x}|_{k=1} = 2q\mu\sqrt{\frac{n}{m}}. \tag{19}$$

Then, let us see the case of computing $\mathbb{E}|\boldsymbol{r}^\top \boldsymbol{x}|_{k>1}$. By (5), $\mathbb{E}|\boldsymbol{r}^\top \boldsymbol{x}|_{k>1}$ is the sum of $\frac{2}{\sqrt{k}}C_k^i p^i q^{k-i}\left\lceil\frac{k-i}{2}\right\rceil C_{k-i}^{\lceil\frac{k-i}{2}\rceil}$ multiplied by $\mu\sqrt{\frac{n}{m}}$. To compute $\frac{2}{\sqrt{k}}C_k^i p^i q^{k-i}\left\lceil\frac{k-i}{2}\right\rceil C_{k-i}^{\lceil\frac{k-i}{2}\rceil}$, we consider separately two cases: $k - i$ is even or odd, as detailed below.

**Case 1:** Suppose $k - i$ is even. We have

$$\begin{aligned}&\frac{2}{\sqrt{k}}C_k^i p^i q^{k-i}\left\lceil\frac{k-i}{2}\right\rceil C_{k-i}^{\lceil\frac{k-i}{2}\rceil}\\ &\leq \frac{1}{\sqrt{k}}C_k^i p^i q^{k-i}(k-i)2^{k-i}\sqrt{\frac{2}{(k-i)\pi}}\\ &\leq \sqrt{\frac{2}{\pi}}C_k^i p^i (2q)^{k-i},\end{aligned} \tag{20}$$

since $C_{2\gamma}^\gamma \leq \frac{2^{2\gamma}}{\sqrt{\gamma\pi}}$, where $\gamma$ is a positive integer (Stănică, 2001).

**Case 2:** Suppose $k - i$ is odd. We have

$$\frac{2}{\sqrt{k}} C_k^i p^i q^{k-i} \left\lceil \frac{k-i}{2} \right\rceil C_{k-i}^{\lceil \frac{k-i}{2} \rceil}$$

$$\leq \frac{1}{\sqrt{k}} C_k^i p^i q^{k-i} (k-i) 2^{k-i} \sqrt{\frac{2}{(k-i-1)\pi}}$$

$$= \sqrt{\frac{2}{\pi}} C_k^i p^i (2q)^{k-i} \frac{k-i}{\sqrt{k(k-i-1)}} \tag{21}$$

Given $k \geq 5$, we further have

$$\frac{k-i}{\sqrt{k(k-i-1)}} < 1 \quad \text{for} \ \ 2 \leq i \leq k-2,$$

and for $i = k-1$ or $k$,

$$\frac{2}{\sqrt{k}} C_k^i p^i q^{k-i} \left\lceil \frac{k-i}{2} \right\rceil C_{k-i}^{\lceil \frac{k-i}{2} \rceil} < \sqrt{\frac{2}{\pi}} C_k^i p^i (2q)^{k-i}.$$

To sum up, when $k - i$ is odd,

$$\frac{2}{\sqrt{k}} C_k^i p^i q^{k-i} \left\lceil \frac{k-i}{2} \right\rceil C_{k-i}^{\lceil \frac{k-i}{2} \rceil}$$

$$\leq \begin{cases} \sqrt{\frac{2}{\pi}} C_k^i p^i (2q)^{k-i}, & k \geq 5, i \geq 2, \\ \frac{2}{\sqrt{k}} C_k^i p^i q^{k-i} (k-i) C_{k-i-1}^{\frac{k-i-1}{2}}, & \text{otherwise.} \end{cases} \tag{22}$$

According to the results (20) and (22) derived in the above two cases, we know that $\mathbb{E}|\boldsymbol{r}^\top \boldsymbol{x}|_{k>1}$ can be computed in terms of two cases, $2 \leq k \leq 4$ and $k \geq 5$. For the case of $2 \leq k \leq 4$, by (5), we have

$$\mathbb{E}|\boldsymbol{r}^\top \boldsymbol{x}| = \begin{cases} \frac{\mu \sqrt{n}}{\sqrt{2m}} (4q^2 + 4pq), & k = 2, \\ \frac{\mu \sqrt{n}}{\sqrt{3m}} (12q^3 + 12pq^2 + 6p^2 q), & k = 3, \\ \frac{\mu \sqrt{n}}{\sqrt{m}} (12q^4 + 24pq^3 + 12p^2 q^2 + 4p^3 q), & k = 4, \end{cases} \tag{23}$$

and for the case of $k \geq 5$, with (20) and (22), we have

$$\mathbb{E}|\boldsymbol{r}^\top \boldsymbol{x}| \leq \mu \sqrt{\frac{2n}{\pi m}} + \mu \sqrt{\frac{n}{m}} (2q)^5 \left( \frac{3\sqrt{5}}{8} - \sqrt{\frac{2}{\pi}} \right). \tag{24}$$

By (19), (23) and (24), we can derive that

$$\mathbb{E}|\boldsymbol{r}^\top \boldsymbol{x}|_{k=1} > \mathbb{E}|\boldsymbol{r}^\top \boldsymbol{x}|_{k>1}$$

holds under the condition of $p \leq 0.188$. Then P1 is proved.

In what follows, we elaborate the proof of (24) by considering two cases of $k$, being even or odd.

**Case 1:** Suppose $k \geq 5$ and $k$ is even. Combining (20) and (22), we have

$$\mathbb{E}|\boldsymbol{r}^\top \boldsymbol{x}| \leq \mu\sqrt{\frac{n}{m}}C_k^1 p(2q)^{k-1}\left(\frac{\sqrt{k}}{2^{k-1}}C_{k-1}^{\frac{k}{2}-1} - \sqrt{\frac{2}{\pi}}\right)$$
$$+ \mu\sqrt{\frac{2n}{\pi m}}\sum_{i=0}^{k} C_k^i p^i (2q)^{k-i}. \tag{25}$$

Denote $h_1(k) = \frac{\sqrt{k}}{2^{k-1}}C_{k-1}^{\frac{k}{2}-1}$. For

$$\frac{h_1(k+2)}{h_1(k)} = \frac{k+1}{\sqrt{k(k+2)}} > 1$$

we have

$$h_1(k) = \frac{\sqrt{k}}{2^{k-1}}C_{k-1}^{\frac{k}{2}-1} \leq \lim_{k\to\infty} h_1(k) = \sqrt{\frac{2}{\pi}}. \tag{26}$$

Then, it follows from (25) and (26) that

$$\mathbb{E}|\boldsymbol{r}^\top \boldsymbol{x}| \leq \mu\sqrt{\frac{2n}{\pi m}}. \tag{27}$$

**Case 2:** Suppose $k \geq 5$ and $k$ is odd. Combining (20) and (22), we have

$$\mathbb{E}|\boldsymbol{r}^\top \boldsymbol{x}| \leq \mu\sqrt{\frac{n}{m}}C_k^0 (2q)^k\left(\frac{\sqrt{k}}{2^{k-1}}C_{k-1}^{\frac{k-1}{2}} - \sqrt{\frac{2}{\pi}}\right)$$
$$+ \mu\sqrt{\frac{2n}{\pi m}}\sum_{i=0}^{k} C_k^i p^i (2q)^{k-i}. \tag{28}$$

Denote $h_2(k) = \frac{\sqrt{k}}{2^{k-1}}C_{k-1}^{\frac{k-1}{2}}$. For

$$\frac{h_2(k+2)}{h_2(k)} = \frac{\sqrt{k(k+2)}}{k+1} < 1$$

we have

$$h_2(k) = \frac{\sqrt{k}}{2^{k-1}}C_{k-1}^{\frac{k-1}{2}} \leq h_2(5) = \frac{\sqrt{5}}{2^4}C_4^2. \tag{29}$$

Then, it follows from (28) and (29) that

$$\mathbb{E}|\boldsymbol{r}^\top \boldsymbol{x}| \leq \mu\sqrt{\frac{2n}{\pi m}} + \mu\sqrt{\frac{n}{m}}(2q)^5\left(\frac{3\sqrt{5}}{8} - \sqrt{\frac{2}{\pi}}\right).$$

**Proof of P2:** For ease of analysis, we first define the function

$$g(\boldsymbol{r}^\top \boldsymbol{x}; k, p) = \frac{\mathbb{E}|\boldsymbol{r}^\top \boldsymbol{x}|_k}{\mu\sqrt{n/m}} = \mathbb{E}\left|\frac{1}{\sqrt{k}}\sum_{i=1}^{k} z_i\right|, \tag{30}$$

where $\{z_i\}$ is independently and identically distributed and $z_i \in \{-1, 0, 1\}$ with probabilities $\{q, p, q\}$. By the Lindeberg-Lévy Central Limit Theorem, we have

$$\frac{1}{\sqrt{k}}\sum_{i=1}^{k} z_i \rightsquigarrow Z, \tag{31}$$

where $Z \sim N(0, 2q)$.

Then based on (24), we have for $k \geq 5$,

$$\mathbb{E}\left|\frac{1}{\sqrt{k}}\sum_{i=1}^{k} z_i\right| \leq \sqrt{\frac{2}{\pi}} + (2q)^5 \left(\frac{3\sqrt{5}}{8} - \sqrt{\frac{2}{\pi}}\right).$$

It means that

$$\lim_{M \to +\infty} \limsup_{k \to +\infty} \mathbb{E}\left[\left|\frac{1}{\sqrt{k}}\sum_{i=1}^{k} z_i\right| \mathbb{1}\left\{\left|\frac{1}{\sqrt{k}}\sum_{i=1}^{k} z_i\right| > M\right\}\right] = 0.$$

Hence, $\left|\frac{1}{\sqrt{k}}\sum_{i=1}^{k} z_i\right|$ is an asymptotically uniformly integrable sequence.

According to Theorem 2.20 in (Van der Vaart, 2000), we obtain

$$\lim_{k \to +\infty} \frac{\sqrt{m}}{\mu\sqrt{n}}\mathbb{E}|\boldsymbol{r}^\top \boldsymbol{x}| = \lim_{k \to +\infty} \mathbb{E}\left|\frac{1}{\sqrt{k}}\sum_{i=1}^{k} z_i\right|$$
$$= \mathbb{E}|Z|$$
$$= 2\sqrt{\frac{q}{\pi}}.$$

Next, let us investigate the error of the above convergence with respect to $k$. Following the definitions and properties described in (30) and (31), we further suppose $t_i = \frac{1}{\sqrt{2q}}z_i$ and $Q \sim N(0, 1)$, and get

$$\left|\frac{\sqrt{m}}{\mu\sqrt{n}}\mathbb{E}|\boldsymbol{r}^\top \boldsymbol{x}| - 2\sqrt{q/\pi}\right|$$
$$= \left|\mathbb{E}\left|\frac{1}{k}\sum_{i=1}^{k} z_i\right| - \mathbb{E}|Z|\right|$$
$$= \sqrt{2q}\left|\mathbb{E}\left|\frac{1}{k}\sum_{i=1}^{k} t_i\right| - \mathbb{E}|Q|\right|$$
$$\leq \sqrt{2q}d_w\left(\mathbb{E}\left|\frac{1}{k}\sum_{i=1}^{k} t_i\right|, \mathbb{E}|Q|\right)$$

where $d_w(\nu, \upsilon)$ denotes the Kolmogorov metric:

$$d_w(\nu, \upsilon) = \sup_{h \in \mathcal{H}}\left|\int h(x)d\nu(x) - \int h(x)d\upsilon(x)\right|,$$
$$\mathcal{H} = \{h : \mathbb{R} \to \mathbb{R} : |h(x) - h(y)| \leq |x - y|\}.$$

By the Theorem 3.2 in (Ross, 2011), since $\{t_i\}$ are i.i.d and $\mathbb{E}t_i = 0$, $\mathbb{E}t_i^2 = 1$, $\mathbb{E}|t_i|^4 < \infty$, we have

$$d_w\left(\mathbb{E}\left|\frac{1}{k}\sum_{i=1}^{k} t_i\right|, \mathbb{E}|Q|\right) \leq \frac{1}{k^{3/2}}\sum_{i=1}^{k}\mathbb{E}|t_i|^3 + \frac{\sqrt{2}}{\sqrt{\pi}k}\sqrt{\sum_{i=1}^{k}\mathbb{E}t_i^4}$$
$$= \frac{1}{\sqrt{2qk}} + \frac{\sqrt{2}}{\sqrt{2q\pi k}},$$

and then

$$\left|\frac{\sqrt{m}}{\mu\sqrt{n}}\mathbb{E}|\boldsymbol{r}^\top \boldsymbol{x}| - 2\sqrt{q/\pi}\right| \leq \frac{\sqrt{\pi} + \sqrt{2}}{\sqrt{\pi k}}.$$

$\square$

### A.2 Proof of Property 1

*Proof.* This problem can be addressed using the Chebyshev's Inequality, which requires us to first derive $\mathbb{E}z$ and $\mathrm{Var}(z)$. Note that $\mathbb{E}z = \mathbb{E}(\frac{1}{m}\sum_{i=1}^{m}|\boldsymbol{r}_i^\top \boldsymbol{x}|) = \mathbb{E}(|\boldsymbol{r}_i^\top \boldsymbol{x}|)$ has been derived in (5). In the sequel, we need to first solve $\mathrm{Var}(z) = \mathbb{E}z^2 - (\mathbb{E}z)^2$, which has

$$\begin{aligned}
\mathbb{E}z^2 &= \mathbb{E}(\frac{1}{m}\sum_{i=1}^{m}|\boldsymbol{r}_i^\top \boldsymbol{x}|)^2 \\
&= \frac{1}{m^2}\mathbb{E}(\sum_{i=1}^{m}|\boldsymbol{r}_i^\top \boldsymbol{x}|^2) + \frac{1}{m^2}\mathbb{E}(\sum_{i\neq j}|\boldsymbol{r}_i^\top \boldsymbol{x}| \cdot |\boldsymbol{r}_j^\top \boldsymbol{x}|) \\
&= \frac{2q\mu^2 n}{m^2} + \frac{m-1}{2m}\mathbb{E}(|\boldsymbol{r}_i^\top \boldsymbol{x}| \cdot |\boldsymbol{r}_j^\top \boldsymbol{x}|).
\end{aligned} \tag{32}$$

For the second term in the above result, it holds

$$\mathbb{E}(|\boldsymbol{r}_i^\top \boldsymbol{x}| \cdot |\boldsymbol{r}_j^\top \boldsymbol{x}|) \leq \mathrm{Var}(|\boldsymbol{r}_i^\top \boldsymbol{x}|) + (\mathbb{E}|\boldsymbol{r}_i^\top \boldsymbol{x}|)^2 = \mathrm{Var}(|\boldsymbol{r}_i^\top \boldsymbol{x}|) + (\mathbb{E}z)^2, \tag{33}$$

by the covariance property

$$\begin{aligned}
\mathrm{Cov}(|\boldsymbol{r}_i^\top \boldsymbol{x}|, |\boldsymbol{r}_j^\top \boldsymbol{x}|) &= \mathbb{E}(|\boldsymbol{r}_i^\top \boldsymbol{x}| \cdot |\boldsymbol{r}_j^\top \boldsymbol{x}|) - \mathbb{E}|\boldsymbol{r}_i^\top \boldsymbol{x}| \cdot \mathbb{E}|\boldsymbol{r}_j^\top \boldsymbol{x}| \\
&= \rho\sqrt{\mathrm{Var}(|\boldsymbol{r}_i^\top \boldsymbol{x}|)} \cdot \sqrt{\mathrm{Var}(|\boldsymbol{r}_j^\top \boldsymbol{x}|)} \\
&= \rho\mathrm{Var}(|\boldsymbol{r}_i^\top \boldsymbol{x}|),
\end{aligned} \tag{34}$$

where $\rho \in (-1, 1)$ is the correlation coefficient.

Substituting (32) into $\mathrm{Var}(z) = \mathbb{E}z^2 - (\mathbb{E}z)^2$, by the inequality (33) and (18), we can derive

$$\begin{aligned}
\mathrm{Var}(z) &\leq \frac{2q\mu^2 n}{m^2} + \frac{m-1}{2m}[\mathrm{Var}(|\boldsymbol{r}_i^\top \boldsymbol{x}|) + (\mathbb{E}z)^2] - (\mathbb{E}z)^2 \\
&= \frac{2q\mu^2 n}{m^2} + \frac{m-1}{2m} \cdot \frac{2q\mu^2 n}{m^2} - (\mathbb{E}z)^2 \\
&= \frac{(m+1)q\mu^2 n}{m^2} - (\mathbb{E}z)^2.
\end{aligned} \tag{35}$$

With the above inequality about $\mathrm{Var}(z)$, we can further explore the condition that holds the desired probability

$$\Pr\{|z - \mathbb{E}z| \leq \varepsilon\} \geq 1 - \delta. \tag{36}$$

By the Chebyshev's Inequality, (36) will be achieved, if $\mathrm{Var}(z)/\varepsilon^2 \leq \delta$; and according to (35), this condition can be satisfied when $\frac{m^2}{m+1} \geq \frac{q\mu^2 n}{\varepsilon^2 \delta}$.

In the above analysis, we consider a random $\boldsymbol{x}$. For a given $\boldsymbol{x}$, the condition of holding (36) can be further relaxed to $m^2 \geq \frac{2q\mu^2 n}{\varepsilon^2 \delta}$, since in this case $|\boldsymbol{r}_i^\top \boldsymbol{x}|$ is independent between different $i \in [m]$, such that $\mathrm{Var}(z)$ changes to be (18) divided by $m$. □

### A.3 Proof of Theorem 2

*Proof.* First, we derive the absolute moment of $z \sim \mathcal{N}(\mu, \sigma^2)$ as

$$\mathbb{E}|z| = \sqrt{\frac{2}{\pi}}\sigma e^{-\frac{\mu^2}{2\sigma^2}} + \mu\left(1 - 2\Phi\left(-\frac{\mu}{\sigma}\right)\right) \tag{37}$$

which will be used in the sequel. With the distributions of $\boldsymbol{r}$ and $\boldsymbol{x}$, we have $|\boldsymbol{r}^\top \boldsymbol{x}| = \sqrt{\frac{n}{mk}} \left| \sum_{i=1}^{k} x_i \right|$. For easier expression, assume $y = \sum_{i=1}^{k} x_i$, then the distribution of $y$ can be expressed as

$$f(y) = \sum_{i=0}^{k} \sum_{j=0}^{k-i} C_k^i C_{k-i}^j p^i q^{k-i} \frac{1}{\sqrt{2\pi k}\sigma} e^{-\frac{(y-(2j+i-s)\mu)^2}{2k\sigma^2}}.$$

Then, by (37) we can derive that

$$\begin{aligned}
\mathbb{E}|\boldsymbol{r}^\top \boldsymbol{x}| &= \sqrt{\frac{n}{mk}} \sum_{i=0}^{k} \sum_{j=0}^{k-i} \left[ C_k^i C_{k-i}^j p^i q^{k-i} \right. \\
&\quad \left. \times \int_{-\infty}^{+\infty} \frac{|y|}{\sqrt{2\pi k}\sigma} e^{-\frac{(y-(2j+i-s)\mu)^2}{2k\sigma^2}} dy \right] \\
&= 2\mu \sqrt{\frac{n}{mk}} \sum_{i=0}^{k} C_k^i p^i q^{k-i} \left\lceil \frac{k-i}{2} \right\rceil C_{k-i}^{\lceil \frac{k-i}{2} \rceil} \\
&\quad - 2\mu \sqrt{\frac{n}{mk}} \sum_{i=0}^{k} C_k^i p^i q^{k-i} \sum_{j=0}^{k-i} C_{k-i}^j \Phi\left( -\frac{|k-i-2j|\mu}{\sqrt{k}\sigma} \right) \\
&\quad + \sigma \sqrt{\frac{2n}{\pi m}} \sum_{i=0}^{k} C_k^i p^i q^{k-i} \sum_{j=0}^{k-i} C_{k-i}^j e^{-\frac{(k-i-2j)^2 \mu^2}{2k\sigma^2}}
\end{aligned}$$

where $\Phi(\cdot)$ is the distribution function of $\mathcal{N}(0,1)$.

The above equation and (19), (23), (24) together lead to

$$\mathbb{E}|\boldsymbol{r}^\top \boldsymbol{x}| \le \mu \sqrt{\frac{n}{m}} + \sigma \sqrt{\frac{2n}{\pi m}}.$$

Next, we can derive the variance of $|\boldsymbol{r}^\top \boldsymbol{x}|$ as

$$\begin{aligned}
\mathrm{Var}(|\boldsymbol{r}^\top \boldsymbol{x}|) &= Var(\boldsymbol{r}^\top \boldsymbol{x}) - \left( \mathbb{E}|\boldsymbol{r}^\top \boldsymbol{x}| \right)^2 \\
&= \frac{n}{m}(\sigma^2 + 2q\mu^2) - \left( \mathbb{E}\|\boldsymbol{r}^\top \boldsymbol{x}\|_1 \right)^2.
\end{aligned}$$

Finally, the convergence of $\frac{\sqrt{m}}{\mu\sqrt{n}}\mathbb{E}|\boldsymbol{r}^\top \boldsymbol{x}|$ shown in (13) and (14) can be derived in a similar way to the proof of P2 in Theorem 1. $\qquad \square$

# B  Numerical validations of Theorems 1 and 2

## B.1  Numerical validation of Theorem 1

### B.1.1  Validating P1 and P2 by directly computing $\mathbb{E}|\mathbf{r}^\top\mathbf{x}|/(\mu\sqrt{n/m})$ with (5)

To more accurately examine the changing trend of $\mathbb{E}|\boldsymbol{r}^\top\boldsymbol{x}|/(\mu\sqrt{n/m})$ against varying matrix sparsity $k$ (derived in P1 and P2), we directly compute the value of $\mathbb{E}|\boldsymbol{r}^\top\boldsymbol{x}|/(\mu\sqrt{n/m})$ by (5). Note that besides the parameter $k$, $\mathbb{E}|\boldsymbol{r}^\top\boldsymbol{x}|/(\mu\sqrt{n/m})$ also involves the parameter $p$, the probability of $x_i = 0$ as specified in (1). So we investigate $\mathbb{E}|\boldsymbol{r}^\top\boldsymbol{x}|/(\mu\sqrt{n/m})$ over $k \in [1, 500]$ for different $p \in (0, 1)$. For brevity, we here only provide the results of $p = 1/3$ and $2/3$ in Figures 5 (a) and (b). The results exhibit two properties similar to those predicted by P1 and P2:

(P3) When $p \le 1/3$, such as the case of $p = 1/3$ shown in Figure 5(a), $\mathbb{E}|\boldsymbol{r}^\top\boldsymbol{x}|/(\mu\sqrt{n/m})$ tends to achieve its maximum value at $k = 1$, but at other larger $k$ when $p > 1/3$, such as the case of $p = 2/3$ illustrated in Figure 5(b). The results indicate that to maximize $\mathbb{E}|\boldsymbol{r}^\top\boldsymbol{x}|/(\mu\sqrt{n/m})$ at $k = 1$, the condition $p \in [0, 1/3)$ is sufficient, which is broader than the theoretical requirement $p \in [0, 0.188)$ derived from P1. Recall that a wider range of $p$ indicates the probability of $x_i = 0$, and a larger $p$ implies a sparser data $\boldsymbol{x}$.

(P4) With the increasing of $k$, as the two cases of $p = 1/3$ and $2/3$ shown in Figures 5(a) and (b), $\mathbb{E}|\boldsymbol{r}^\top\boldsymbol{x}|/(\mu\sqrt{n/m})$ tends to converge to the limit value $2\sqrt{q/\pi}$ derived in (7), where $q = (1-p)/2$. Furthermore, it can be seen that small convergence errors will be achieved, when $k$ is very small, typically in the range of a few tens. For instance, in Figure 6(a) we derive the convergence error rates as defined in (9), which give the values close to zero when $k \ge 20$ and $p$ is relatively small.

In the analysis of the expected value $\mathbb{E}|\boldsymbol{r}^\top\boldsymbol{x}|$, the influence of the variance $\mathrm{Var}(|\boldsymbol{r}^\top\boldsymbol{x}|)$ in (6) should be considered. Statistically, a lower variance $\mathrm{Var}(|\boldsymbol{r}^\top\boldsymbol{x}|)$ indicates a higher probability that the actual distance $|\boldsymbol{r}^\top\boldsymbol{x}|$ of a single matrix closely approximates its expected value $\mathbb{E}|\boldsymbol{r}^\top\boldsymbol{x}|$. Also, this implies a higher consistence between theoretical and practical results. By computing (6), we observe a trend similar to $\mathbb{E}|\boldsymbol{r}^\top\boldsymbol{x}|$: as $k$ increases, $\mathrm{Var}(|\boldsymbol{r}^\top\boldsymbol{x}|)$ tends to quickly converge to a constant value. This suggests that $\mathrm{Var}(|\boldsymbol{r}^\top\boldsymbol{x}|)$ varies minimally across different $k$ values. Therefore, the probability of $|\boldsymbol{r}^\top\boldsymbol{x}|$ approximating $\mathbb{E}|\boldsymbol{r}^\top\boldsymbol{x}|$ remains consistent for different $k$, thereby ensuring $|\boldsymbol{r}^\top\boldsymbol{x}|$ to exhibit similar changing trends with $\mathbb{E}|\boldsymbol{r}^\top\boldsymbol{x}|$ when varying $k$.

### B.1.2  Validating P1 and P2 by statistically estimating $\mathbb{E}|\mathbf{r}^\top\mathbf{x}|/(\mu\sqrt{n/m})$ with synthetic data

To verify the correctness of Theorem 1, including the expression (5) of $\mathbb{E}|\boldsymbol{r}^\top\boldsymbol{x}|$ and its two properties P1 and P2, we here estimate the expectation value $\mathbb{E}|\boldsymbol{r}^\top\boldsymbol{x}|/(\mu\sqrt{n/m})$ (against varying $k$) by averaging over the statistically generated samples of $\boldsymbol{r}$ and $\boldsymbol{x}$. If the theorem results are correct, the statistical simulation results should be consistent with the numerical computation results P3 and P4 (derived by Theorem 1). The simulation is introduced as follows. First, we randomly generate $10^6$ pairs of $\boldsymbol{r}$ and $\boldsymbol{x}$ from their respective distributions, i.e. $\boldsymbol{r} \in \{0, \pm\sqrt{\frac{n}{mk}}\}^n$ with $k$ nonzero entries randomly distributed, and $\boldsymbol{x}$ with i.i.d. $x_i \sim \mathcal{T}(\mu, p, q)$. Then, the average value of $|\boldsymbol{r}^\top\boldsymbol{x}|/(\mu\sqrt{n/m})$ is derived as the final estimate of $\mathbb{E}|\boldsymbol{r}^\top\boldsymbol{x}|/(\mu\sqrt{n/m})$. The parameters for the distributions of $\boldsymbol{r}$ and $\boldsymbol{x}$ are set as follows: $m = 1$, $n = 10^4$, $\mu = 1$, and $p = 1/3$ or $2/3$. The data dimension $n = 10^4$ allows us to increase $k$ from 1 to $10^4$. The average value of $|\boldsymbol{r}^\top\boldsymbol{x}|/(\mu\sqrt{n/m})$ at each $k$ is provided in Figures 5(c) and (d), respectively for the cases of $p = 1/3$ and $2/3$. Note that the choices of $m$, $n$ and $\mu$ will not affect the changing trend of $\mathbb{E}|\boldsymbol{r}^\top\boldsymbol{x}|/(\mu\sqrt{n/m})$ against $k$. Comparing the numerical computation results with the simulation results presented in Figure 5, specifically contrasting (a) vs. (c) and (b) vs. (d), it is seen that both sets of results exhibit similar trends in the variation of $\mathbb{E}|\boldsymbol{r}^\top\boldsymbol{x}|/(\mu\sqrt{n/m})$. The similarity between them validates Theorem 1, as well as the numerical computation results P3 and P4.

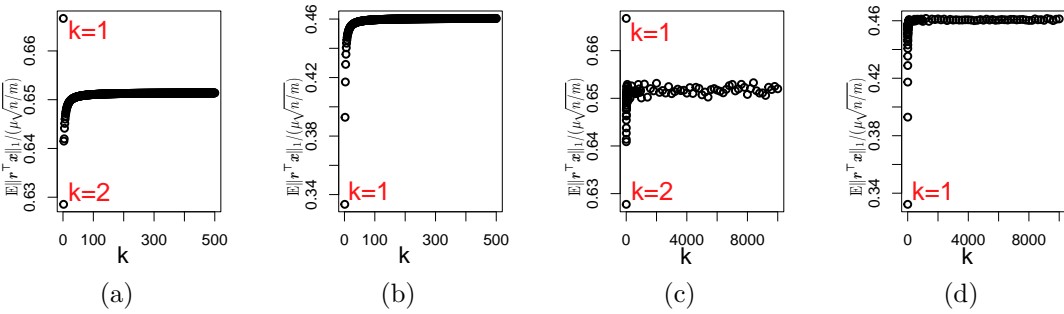

Figure 5: The value of $\mathbb{E}|\boldsymbol{r}^\top \boldsymbol{x}|/(\mu\sqrt{n/m})$ calculated by (5) with $p = 1/3$ (a) and $p = 2/3$ (b), and estimated by statistical simulation with $p = 1/3$ (c) and $p = 2/3$ (d), provided $x_i \sim \mathcal{T}(\mu, p, q)$, $\mu = 1$.

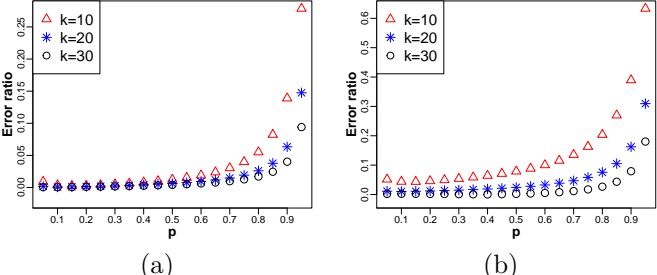

Figure 6: The convergence error rates of three different $k \in \{10, 20, 30\}$ over varying $p$ are derived for ternary data (a) and Gaussian-noised ternary data (b), by computing the inequality of $\eta$ respectively in (9) and (15).

## B.2 Numerical validation of Theorem 2

### B.2.1 Validating (13) by directly computing $\mathbb{E}|\mathbf{r}^\top \mathbf{x}|/(\mu\sqrt{n/m})$ with (10)

In this part, we directly compute the value of $\mathbb{E}|\boldsymbol{r}^\top \boldsymbol{x}|/(\mu\sqrt{n/m})$ by (10). Note that $\mathbb{E}|\boldsymbol{r}^\top \boldsymbol{x}|/(\mu\sqrt{n/m})$ involves four parameters: $k$, $p$, $\mu$, and $\sigma$. In computing (10), we fix $\mu = 1$ and vary other parameters in the ranges of $\sigma/\mu \in (0, 1/3)$, $p \in (0, 1)$ and $k \in [1, 500]$. Here, we upper bound the value range of $\sigma/\mu$ by $1/3$ for easy simulation. Empirically, the changing trend of $\mathbb{E}|\boldsymbol{r}^\top \boldsymbol{x}|/(\mu\sqrt{n/m})$ is not sensitive to the varying of $\sigma/\mu$, but sensitive to $p$, i.e. the probability of each entry $x_i$ of the data difference $\boldsymbol{x}$ taking zero value, as specified in (2). In Figures 7(a) and (b), we provide two typical results of $p = 1/2$ and $2/3$, and observe two properties similar to the previous P3 and P4:

(P5) When $p \leq 1/2$, such as the case of $p = 1/2$ and $\sigma/\mu = 1/3$ shown in Figure 7(a), $\mathbb{E}|\boldsymbol{r}^\top \boldsymbol{x}|/\mu\sqrt{n/m}$ tends to obtain its maximum at $k = 1$, but at other larger $k$ when $p > 1/2$, such as the case of $p = 2/3$ and $\sigma/\mu = 1/3$ shown in Figure 7(b). It can be seen that the upper bound of $p$ obtained here for Gaussian mixture data is relaxed from $2/3$ to $1/2$ compared to the bound derived in P3 for two-point distributed data. This implies a wider range of data distributions that enable obtaining the maximum $\mathbb{E}|\boldsymbol{r}^\top \boldsymbol{x}|/\mu\sqrt{n/m}$ at $k = 1$.

(P6) With the increasing of $k$, as the two results shown in Figure 7(a) and (b), $\mathbb{E}|\boldsymbol{r}^\top \boldsymbol{x}|/(\mu\sqrt{n/m})$ converges to the limit value derived in (13). Similarly to the convergence discussed in P4 for two-point distributed data, the convergence error ratio defined in (15) can approach zero with small $k$, such as $k = 20$ shown in Figure 6(b), especially when $p$ is relatively small, namely the original data $\boldsymbol{x}$ having relatively dense distributions.

For P5 and P6, their similarity to P3 and P4 is not surprising, since the ternary discrete distribution $x_i \sim \mathcal{T}(\mu, p, q)$ can be viewed as an extreme case of the three-component Gaussian mixture $x_i \sim \mathcal{M}(\mu, \sigma^2, p, q)$

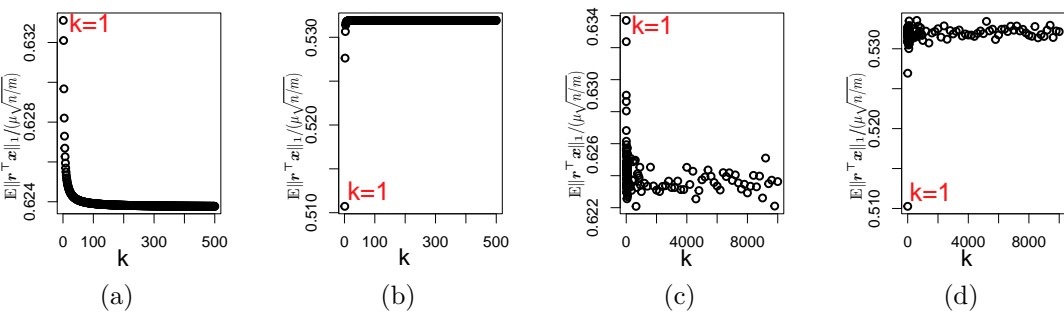

Figure 7: The value of $\mathbb{E}|\boldsymbol{r}^\top\boldsymbol{x}|/\sqrt{n/m}$ calculated by (10) with $p = 1/2$ (a) and $p = 2/3$ (b), and estimated by statistical simulation with $p = 1/2$ (c) and $p = 2/3$ (d), provided $x_i \sim \mathcal{M}(p, q, \mu, \sigma^2)$, $\mu = 1$ and $\sigma = 1/3$.

with $\sigma \to 0$. Thanks to the good generalizability of Gaussian mixture models, as will be seen in our experiments, the two properties analyzed above apply to a variety of real-world data.

### B.2.2  Validating (13) by statistically estimating $\mathbb{E}|\mathbf{r}^\top\mathbf{x}|/(\mu\sqrt{n/m})$ with synthetic data

Similarly as in Section B.1.2, we here verify the correctness of Theorem 2, including the expression (10) of $\mathbb{E}|\boldsymbol{r}^\top\boldsymbol{x}|$ and its convergence (13) by performing statistical simulations on $\boldsymbol{x}$ and $\boldsymbol{r}$. The simulation results should agree with the numerical computation results P5 and P6, if the theorem is correct. In the simulation, we estimate the value of $\mathbb{E}|\boldsymbol{r}^\top\boldsymbol{x}|/\sqrt{n/m}$ by drawing $10^6$ pairs of $\boldsymbol{x}$ and $\boldsymbol{r}$ from their respective distributions and then computing the average of $|\boldsymbol{r}^\top\boldsymbol{x}|_1/\sqrt{n/m}$ as the estimate. The parameters of the distributions of $\boldsymbol{x}$ and $\boldsymbol{r}$ are set as follows: $m = 1$, $n = 10000$, $\mu = 1$, $\sigma = 1/3$ and $p = 1/2$ or $2/3$. The data dimension $n = 10000$ allows $k$ to vary between 1 and 10000. The average value of $|\boldsymbol{r}^\top\boldsymbol{x}|/\sqrt{n/m}$ at each $k$ is presented in Figures 7(c) and (d), with $p = 1/2$ and $2/3$, respectively. Comparing the numerical computation results and the simulation results shown in Figure 7, specifically contrasting (a) vs. (c) and (b) vs. (d), it can be seen that two kinds of results are roughly consistent with each other. The consistency validates Theorem 2, as well as the numerical computation results P5 and P6.

