# OpenReview forum: "The Sparse Matrix-Based Random Projection: A Mean Absolute Deviation Analysis for Sparse Ternary Data"
_TMLR — Rejected by TMLR_

### Review · Reviewer_nZ33 · 2026-02-01

**Summary Of Contributions:**

This paper studies the projection of $\{0, \pm\mu\}$ sparse ternary data using sparse $\{0, \pm1\}$ matrices by analyzing the mean absolute deviation (MAD). The authors derive closed-form MAD expressions for sparse ternary data and data with additive Gaussian noise. As corollaries, the authors argue that (i) for sufficiently dense data, MAD is maximized when each row of the sparse random matrix contains only one nonzero entry, and (ii) MAD converges rapidly to a stable level as the number of nonzero entries in the random projection matrix increases. Experiments confirm that extremely sparse projections indeed obtain favorable classification performance, suggesting the potential of substantial computational savings via random projection.

**Audience:**

No

**Audience Explanation:**

The findings of this paper is limited to the analysis of MAD, which as far as I know is not a popular metric to study. Moreover, the derivations, while rigorous, don't seem to contain proof techniques that could be useful elsewhere.

**Claims And Evidence:**

No

**Claims Explanation:**

1. The authors attempt to explain the good classification performance of projected data using high MAD, but I find that unsatisfactory. The only reasoning provided is heuristic arguments like *"high degree of dispersion indicates that the projected points retain more of the variation inherent in the original data points (Jolliffe & Cadima, 2016; Singh, 2023), thereby favorable for downstream classification tasks"*, with little theoretical under pinning.
2. I also find the authors' treatment of additive Gaussian noise confusing, because their Equation (2) is adding Gaussian noise to the quantized data, instead of quantizing data with additive Gaussian noise.
3. The experiments are problematic because
    1. the authors didn't show the relationship between MAD and classification accuracy, which is crucial to support their argument that *"a larger MAD(z) probably yields better classification performance"*, and
    2. statements such as *"Classification is conducted both on the original full-precision features and their ternary quantization counterparts, which serve to validate Theorems 2 and 1, respectively."* equates the original full-precision features with Gaussian-corrupted quantized data, which isn't realistic.

**Requested Changes:**

1. Instead of additive Gaussian noise, it's perhaps more realistic to consider symbol flips ([BSC](https://en.wikipedia.org/wiki/Binary_symmetric_channel)) or erasures ([BEC](https://en.wikipedia.org/wiki/Binary_erasure_channel)) in this quantized setting. Note that this is in some sense equivalent to quantizing data with additive Gaussian noise.
1. Consider non-symmetric ternary data, e.g., a model where $-\mu$ and $\mu$ doesn't appear with the same probability. It's

---

> ### Author Response · Authors · 2026-02-01
> **Response to Reviewer nZ33**
>
> Dear Reviewer nZ33,
>
> We sincerely appreciate your time in reviewing our manuscript and providing us with valuable comments. We have addressed all your concerns and provided detailed explanations for each one below.
>
> **Comment 1:** The authors are using high MAD to explain the good classification performance of projected data, which I find unsatisfactory. The only reasoning provided is heuristic arguments like "high degree of dispersion indicates that the projected points retain more of the variation inherent in the original data points (Jolliffe & Cadima, 2016; Singh, 2023), thereby favorable for downstream classification tasks", which little theoretical under pinning.
>
>
> **Response 1:** We would like to clarify that the mean absolute deviation (MAD) is essentially a robust variant of the variance (or called variation/dispersion/spread) analysis adopted in principal component analysis (PCA),  which replaces the L2 norm distance metric used in PCA with the L1 norm [r1, r2]. Consequently, the projection analysis based on MAD is also known as L1-PCA [r1, r2] or robust PCA  (McCoy
> & Tropp, 2011). This method  has been extensively studied for improving the  robustness of PCA, as evidenced by the numerous citations of [r1].
>
> **[r1]** N. Kwak, "Principal Component Analysis Based on L1-Norm Maximization," in IEEE Transactions on Pattern Analysis and Machine Intelligence, vol. 30, no. 9, pp. 1672-1680, Sept. 2008.
>
> **[r2]** Deyu Meng, Qian Zhao, and Zongben Xu. Improve robustness of sparse PCA by l1-norm maximization.
> Pattern Recognition, 45(1):487–497, 2012.
>
>
>
> As the reviewer  noted, exploring the theoretical relationship between classification accuracy and the variance of projected data points (whether based on the L2 norm in PCA or the L1 norm in MAD) is indeed intriguing. This should form the foundation that supports the widespread use of PCA in classification for dimensionality reduction and feature extraction. Unfortunately, to the best of our knowledge, no rigorous theoretical analysis currently exists to establish this connection, despite PCA’s long history. The prevailing explanation is that greater variance in projected data points tends to capture more variations present in the original data, thereby improving classification performance (Jolliffe & Cadima, 2016; [r1,r2]).
>
>
> Although rigorous theoretical support is lacking, the idea that larger variance improves classification has been widely validated through extensive research and practical applications of PCA. Many studies that maximize variance via PCA (or MAD [r1, r2]) have achieved favorable classification results. This empirical success has led to the widespread use of PCA in machine learning for dimensionality reduction and feature extraction, even in the absence of strict theoretical guarantees.
>
>
> Overall, as a robust variant of PCA, i.e., L1-norm-based PCA，MAD is an important method for analyzing linear projections. The notion that larger MAD tends to yield higher classification accuracy has been widely recognized in the PCA literature (Jolliffe & Cadima, 2016; [r1, r2]), despite the lack of strict theoretical guarantees. As supporting evidence, our classification experimental results are consistent with our MAD-based analysis.
>
>
>
> **Comment 2:** I also find the authors' treatment of additive Gaussian noise confusing, because they appear to be adding Gaussian noise to the quantized data, instead of quantizing data with additive Gaussian noise.
>
> **Response 2:** Thank you for pointing out this issue. Indeed, we  propose to analyze the ternary data with additive Gaussian noise, rather than the ternary quantization of the original data that has been corrupted by Gaussian noise. Our choice  is motivated by two main reasons.
>
> First, the core focus of this paper is the random projection of ternary data, rather than the ternary quantization of full-precision data.
>
> Second, we aim to investigate whether our MAD-based analysis can be extended to unquantized real-valued data, in addition to ternary-quantized data. For this purpose, we study the scenario where Gaussian noise is added to ternary data. Given that directly evaluating the classification performance of ternary data with Gaussian noise lacks practical significance, in our experiments we roughly treat the original features (from which the ternary data are generated) as a proxy of the noisy ternary data, and evaluate their classification performance. The experimental results demonstrate that our MAD-based classification analysis can indeed be extended to the original full-precision features, validating the generalizability of our theoretical findings.

---

> ### Author Response · Authors · 2026-02-01
>
> **Comment 3:** The experiments are also problematic because
>
> 1) the authors didn't show the relationship between MAD and classification accuracy, which is crucial to support their argument that "a larger MAD(z) probably yields better classification performance", and
>
> 2) statements such as "Classification is conducted both on the original full-precision features and their ternary quantization counterparts, which serve to validate Theorems 2 and 1, respectively." equates the original full-precision features with Gaussian-corrupted quantized data, which isn't realistic.
>
> **Response 3:**  The first question has been addressed in Response 2. We elaborate on the second question as follows. First, we would like to clarify that the correctness of Theorems 1 and 2, which pertain to MAD values (not classification accuracy), is validated through numerical simulations in Appendix B, rather than classification experiments. Classification experiments are specifically designed to verify whether classification performance aligns with our MAD-based analysis.
>
> As replied in Response 2, we acknowledge that modeling original full-precision features using their Gaussian-noised ternary quantization  is not mathematically rigorous.  In our experiments, we use the original features (from which the ternary data are generated) as a proxy for the noisy ternary data and evaluate their classification performance. This is because the classification performance of ternary data with Gaussian noise lacks practical significance, even though it aligns with our MAD analysis.  Experimental results demonstrate that the classification performance of the original full-precision features is consistent with our MAD-based classification analysis ,  validating the generalizability and practical value of our theoretical findings (not limited to ternary data).
>
> **Comment 4:** Instead of additive Gaussian noise, it's perhaps more realistic to consider symbol flips (BSC) or erasures (BEC) in this quantized setting.
>
> **Response 4:**  As clarified in Response 2, we propose  Gaussian-noised ternary data   to approximate real-valued data and investigate their MAD (not limited to ternary data), rather than to examine the impact of Gaussian noise on ternary quantization. In other words, the main  goal of this paper is to investigate the random projection of *pre-existing* ternary data, rather than the generation  of  (or  noise on)  these *ternary* data. Thus, discussions regarding ternary quantization and quantization noise models, such as BSC and BEC, fall outside the scope of this work.
>
> **Comment 5:**   Consider non-symmetric ternary data, e.g., a model where $-\mu$ and $\mu$ doesn't appear with the same probability.
>
> **Response 5:**  Thank you for your suggestion. In theoretical research, achieving universality is a key objective. Current theoretical work typically models real-world data features using symmetric distributions, such as the Gaussian (dense) and Laplace (sparse) distributions. This makes the symmetry assumption statistically more general and meaningful relative to non-symmetric cases.
>
> Furthermore, it is  worth noting that the non-symmetric case can be analyzed using the same approach as the symmetric case. However, the non-symmetric case requires  an additional parameter to describe the different distributions of $\pm \mu$. This makes the analysis  and results overly complex, difficult to describe and understand intuitively, and thus lacking practical utility.

---

### Review · Reviewer_QM2R · 2026-02-19

**Summary Of Contributions:**

This paper analyzes sparse random project methods, in two disconnected ways:
  - theoretical bounds on Mean Absolute Deviation (MAD)
  - experiments on KNN classification of a few data sets.

**Audience:**

No

**Audience Explanation:**

It is not clear why the MAD is relevant of better than common distortion bounds.  The motivation mentions some old papers related to PCA, but I was not convinced.

Experiments of classification under random projection is KNN classification is maybe of mild interest.  But with fast practical algorithms for high-dimensional NN search, this seems less relevant nowadays.

The paper does not discuss well-known theoretical results in related topics, such as:
  - subGaussian analysis:  On variants of the Johnson–Lindenstrauss lemma.  Matousek 2008
  - sparse JL:  Low-rank approximation and regression in input sparsity time.  Clarkson+Woodruff JACM 2017
  - and extensions:  OSNAP: Faster numerical linear algebra algorithms via sparser subspace embeddings.  Nelson+Nguyen STOC 2013.

The important take-away from all of these works is that random project bounds are somewhat interesting by themselves, but relevant to ML when they have implications in analysis tasks (like regression).  This paper *does not* answer the interesting question of: why are MAD bounds useful for KNN Classification?  Or why are MAD bounds relevant for other ML tasks.

**Claims And Evidence:**

Yes

**Claims Explanation:**

Note:  I had an incorrect statement about the derivation of MAD used in the analysis.  It has been removed.

The analysis looks technical, but straight-forward (the main parts are left to the Appendix with no insight otherwise).




On page 4 in **Regarding P1**, the paper claims "$x$ needs to exhibit a sufficiently dense distribution" as a consequence of a bound with premise p <= 0.188  (with p the probability of getting a 0 in the ternary {0,-1,+1} distribution).  But that is not how to read this bound. It says if p is small you can claim something, but it does not prohibit something good from happening with small p.

**Requested Changes:**

There are 2 things that might help in a short rebuttal period -- but I am skeptical the paper can get to a suitable state for publication in a small revision, and may be better to rework and resubmit.


 1.  Connect the MAD bounds derived to the experiments.  Why is MAD the right thing to analyze for KNN classifiers?  Is it?

 2.  Fix the issue with P1 on page 4.

---

> ### Author Response · Authors · 2026-02-22
> **Response to Reviewer QM2R**
>
> Dear Reviewer QM2R,
>
> We sincerely appreciate your time spent reviewing our manuscript and the valuable comments you have provided. We have   addressed your concerns one by one below. If you have any further questions, we welcome further discussion.
>
> **Comment 1:** Fix explanation of MAD with a better derivation of the form analyzed in Section 2.3.
>
> **Response 1:** Please note that there is a derivation for the form of MAD in the last paragraph of Section 2.3.  We have rechecked the derivation, and it is correct.
>
> **Comment 2:**  Connect the MAD bounds derived to the experiments. Why is MAD the right thing to analyze for KNN classifiers? Is it?
>
> **Response 2:**
> 1) **Connection between MAD and PCA.**  It is worth noting that the mean absolute deviation (MAD) is essentially a robust variant of the variance (or called variation/dispersion/spread) analysis adopted in principal component analysis (PCA)   [r1], which replaces the L2 norm distance metric $E||z-Ez||_2$ used in PCA with the L1 norm metric $E||z-Ez||_1$, $z=Rx$. Consequently, the projection analysis based on MAD is also known as L1-PCA [r1] or robust PCA (McCoy & Tropp, 2011), which has been widely studied in machine learning for dimensionality reduction and feature extraction, as evidenced by the numerous citations of [r1].
>
>      [r1] N. Kwak, "Principal Component Analysis Based on L1-Norm Maximization," in IEEE Transactions on Pattern Analysis and Machine Intelligence, vol. 30, no. 9, pp. 1672-1680, Sept. 2008.
>
> 2) **Connection between MAD and Classification performance.** In PCA, a projection $z$ with a larger variance value $E||z-Ez||_2$ is typically expected to yield better classification performance,  as it captures more variations in the original data. This principle underpins PCA and supports its wide applications in various classification tasks. Similarly, a projection $z$ with a larger MAD value $E||z-Ez||_1$ is also expected to achieve better classification performance [r1].
>
> 3) **Connection between MAD and KNN.**  Our MAD analysis aligns with classification results not only for the KNN classifier but also for other  classifiers such as SVM. This is because our MAD analysis focuses on data distributions rather than specific  classifiers.  In this paper, we present results using the naive KNN mainly because it can directly reflect the discriminability of the data without introducing additional feature selection or enhancement operations.
>
>
>
>
> **Comment 3:**  It is not clear why the MAD is relevant of better than common distortion bounds. The motivation mentions some old papers related to PCA, but I was not convinced.
>
> **Response 3:**
>
> 1) **Connection between random projection and PCA.** Both random projection and PCA are unsupervised linear projection methods, used to reduce the complexity of downstream classification tasks. The main difference is that traditional random projection analysis focuses on preserving pairwise distances between the original data (simply called distance preservation), whereas PCA aims to maximize the variance of the projected data to capture more variations in the original data.
>
> 2) **Limitation of traditional distance preservation analysis.**  However, it is worth noting that the distance preservation property of projected data (as studied in random projection) cannot directly or accurately reflect classification performance, since classification depends on the discriminability of the data. As shown in our paper, classification performance does not necessarily become worse as the projection matrix becomes sparser, with weaker distance preservation (Li et al., 2006).
>
> 3) **Advantage of MAD over distance preservation.** For the above reason, we propose using MAD (L1-PCA) instead of distance preservation to analyze random projections. Our experiments demonstrate that the varying trend of MAD caused by changes in matrix sparsity, aligns closely with the observed variations in classification performance. This empirical consistency validates the effectiveness of our MAD-based analysis.
>
>     Overall, our research demonstrates that MAD (L1-PCA) outperforms distance preservation in estimating matrix sparsity for desired classification performance. This discrepancy may stem from the fact that PCA prioritizes global variance/dispersion, which offers a coarse characterization of the data distribution but allows accurate measuring/computing. In contrast, while distance preservation property provides a more refined structural description, its accuracy (namely the probability that ensures distance errors within given  bounds) degrades rapidly as the compression ratio $m/n$ of random matrices decreases, leading to a mismatch between theoretical results and practical performance.

---

> > ### Comment · Reviewer_QM2R · 2026-02-22
> >
> > You are totally correct on the derivation of MAD.  I apologize for my poor reading!
> >
> > I still do not see a direct connection (e.g., can you prove anything) about KNN classification (or any classification method) under this form of sparse regression.  Without that the two parts feel disconnected.

---

> ### Author Response · Authors · 2026-02-22
>
> **Comment 4:**  On page 4 in Regarding P1, the paper claims " needs to exhibit a sufficiently dense distribution" as a consequence of a bound with premise p <= 0.188 (with p the probability of getting a 0 in the ternary {0,-1,+1} distribution). But that is not how to read this bound. It says if p is small you can claim something, but it does not prohibit something good from happening with small p.
>
> **Response 4:** Thank you for pointing out this issue with our phrasing, and we will revise it accordingly. As you noted, the desired property P2 holds even when the constraint $p<=0.188$ is not satisfied.
>
> **Comment 5:** The paper does not discuss well-known theoretical results in related topics, such as the three references listed above.
>
> **Response 5:** Thank you for highlighting these three references. We note that the works by Clarkson and Woodruff (JACM 2017) and Nelson and Nguyen (STOC 2013) focus on subspace embeddings, which assume data originates from closed linear subspaces. However, their theoretical frameworks cannot be directly extended to our study of $\\{0,\pm1\\}$ ternary data, as ternary vectors do not satisfy the closure property under linear combinations, e.g., the sum of two ternary vectors is not guaranteed to remain ternary.
>
> The work by Matousek (2008) investigated the influence of matrix sparsity on distance preservation in Theorem 4.1, deriving sparsity bounds that depend on two undetermined parameters, $C$ and $\alpha$.  In contrast, our bounds ( derived in (9) and (15)) feature explicit, computable parameters, offering greater practical applicability.
>
> Moreover, it is important to note that a direct comparison between our matrix sparsity bounds and those in prior works (e.g., Matousek 2008) is not strictly valid, as our bounds are optimized for maximizing MAD (L1-PCA), whereas existing bounds focus on distance preservation. The key advantage of our MAD-based analysis is that it provides sparsity bounds that align more closely with empirical classification performance, making them more relevant for classification-related applications.
>
> **Comment 6:** The logical coherence and notable contributions of this work
>
> **Response 6:** Our study demonstrates both theoretical rigor and practical significance through the following contributions:
>
> 1)	**Theoretical foundation.** We establish a rigorous connection between the sparsity of random matrices and MAD (L1-PCA), supported by theoretical proofs and numerical validation.
>
> 2)	**Link to classification.** The relationship between MAD (L1-PCA) and classification performance, where larger variances/deviations typically yield better feature extraction and classification outcomes, is widely recognized in the extensive literature on PCA. This principle constitutes a fundamental basis for the wide applications of PCA in various classification tasks.
>
> 3)	**Empirical validation.** Our experiments have sufficiently validated the relationship between matrix sparsity and classification performance, as implied by our MAD analysis, confirming the practical relevance of our theoretical findings.

---

> > ### Comment · Reviewer_QM2R · 2026-02-22
> >
> > I agree that the Matousek and sparse JL (via CountSketch) approaches are not directly related to the specific bounds, but:
> >
> >  - the Matousek paper provides the most general bounds for standard distortion bounds for {-1, 0, +1} and related forms.
> >
> >  -  the CountSketch papers (Clarkson+Woodruff & Nelson+Nguyen) show how to connect sparse analysis directly to ML results (in that case regression).  That is their bounds lead directly to claims about the accuracy of regression.

---

> ### Author Response · Authors · 2026-02-22
> **Response to the connection between MAD and classification**
>
> **Comment 7:** You are totally correct on the derivation of MAD. I apologize for my poor reading!  I still do not see a direct connection (e.g., can you prove anything) about KNN classification (or any classification method) under this form of sparse regression. Without that the two parts feel disconnected.
>
>
>
>
> **Response 7:** Thank you for your further comments. In this paper, we focus on estimating the matrix sparsity that maximizes the MAD of random projections, with the expectation (and empirical validation) that such sparsity also enhances classification performance of projections. This expectation is grounded in the observation that higher MAD values tend to yield better projection features for classification (N. Kwak, 2008).
>
> More precisely, as clarified in our earlier Responses 2 and 3, PCA research has long established that a projection
> $z$ with higher variance  $E||z-Ez||_2$ (Jolliff, 2016) or higher MAD  $E||z-Ez||_1$ (Kwak, 2008) tends to capture more variability in the original data, making it more suitable for downstream classification tasks. This property has been widely demonstrated through PCA’s extensive applications in various classification problems, including its pioneering use in face recognition (Turk & Pentland, 1991) and subsequent extensions.

---

> ### Author Response · Authors · 2026-02-22
> **Response to discussion on related works**
>
> **Comment 8:** I agree that the Matousek and sparse JL (via CountSketch) approaches are not directly related to the specific bounds, but: 1)  the Matousek paper provides the most general bounds for standard distortion bounds for {-1, 0, +1} and related forms. 2) the CountSketch papers (Clarkson+Woodruff & Nelson+Nguyen) show how to connect sparse analysis directly to ML results (in that case regression). That is their bounds lead directly to claims about the accuracy of regression.
>
> **Response 8:** This question is interesting and important. Note that the sparsity bounds in the referenced works are derived under the constraint of *distance (or norm) preservation*, which provide performance guarantees for several regression and low-rank approximation problems. However, these bounds *cannot* provide  perfect performance guarantees for classification tasks. This limitation arises from the fact that  classification performance depends on the discriminative power (i.e., inter-class separability versus intra-class compactness) of the projected data, rather than their structural similarity to the original data. In other words, projections that extract key discriminative features, even if they exhibit drastically different structures from the original data, can achieve superior classification performance.
>
> Our research demonstrates that MAD (L1-PCA) outperforms distance preservation in estimating matrix sparsity for desired classification performance. Besides the fact that distance preservation is not inherently necessary for classification,   as discussed in Response 3, another key reason is  that PCA prioritizes global variance/dispersion, which offers a coarse characterization of the data distribution but allows accurate measuring/computing. In contrast, while distance preservation property provides a more refined structural description, its accuracy (namely the probability that ensures the distance error within given bounds) degrades rapidly as the compression ratio $m/n$ of random matrices decreases, leading to a mismatch between theoretical results and practical performance.

---

> > ### Comment · Reviewer_QM2R · 2026-03-11
> >
> > I am still not getting the argument for the theoretical bounds on MAD connected to empirical bounds on k-NN classification.  This passage:
> >
> > > In PCA, a projection  $z$ with a larger variance value $E\|z - Ez\|_2$ is typically expected to yield better classification performance, as it captures more variations in the original data. This principle underpins PCA and supports its wide applications in various classification tasks. Similarly, a projection $z$ with a larger MAD value $E \|z - Ez\|_1$ is also expected to achieve better classification performance [r1].
> >
> > just restated your claim.  It does not explain why it is true.
> >
> > There are elements of a paper here that could get into TMLR, but the pieces do not fit.  The MAD bound may be interesting in direct dimensionality reduction applications (e.g., I guess for Vis?).  Or you could try to connect the mathematical bounds on MAD to **mathematically proven** bounds on classification.  But the connection of theory on MAD for empirical work on k-NN classifiers still seems odd to me.  JL bounds which preserve all distance up to (1+eps) seems a fit for k-NN bounds, not the bounds you provide.

---

> ### Author Response · Authors · 2026-04-10
> **Response to connection between MAD and classification**
>
> **Comment 9:**  I am still not getting the argument for the theoretical bounds on MAD connected to empirical bounds on k-NN classification. There are elements of a paper here that could get into TMLR, but the pieces do not fit. The MAD bound may be interesting in direct dimensionality reduction applications (e.g., I guess for Vis?). Or you could try to connect the mathematical bounds on MAD to mathematically proven bounds on classification.
>
> **Response 9:**  Thank you for your further question. As noted earlier, the dispersion of projected points, measured by variance in $\ell_2$-metric and by MAD in $\ell_1$-metric, is at the core of PCA analysis.  Generally, projections with higher dispersion are considered to generate better features for classification (Turk & Pentland, 1991), due to retaining more variability/information from the original data ((Jolliffe & Cadima, 2016). The relevant theoretical basis is presented as follows.
>
> In the early studies of  PCA  (Rao, 1964; Fukunaga, 2013: Equation 10.4),    it is established that the total scatter matrix $S_t$​ can be decomposed into the between-class scatter matrix $S_b$​ and the within-class scatter matrix $S_w​$, i.e., $S_t​=S_b​+S_w$​. Based on this fundamental decomposition, as noted in Jolliffe (2002, page 202), it is straightforward to conclude that  when the between-class variance $x^TS_b​x$  (of the projected points) is sufficiently larger than the within-class variance $x^TS_w​x$ (of the projected points), namely  the data exhibit sufficiently high discriminability, the total variance $x^TS_t​x$ (of the projected points) will be dominated by the between-class variance $x^TS_b​x$. In this case,  increasing the total variance $x^TS_t​x$ will directly lead to an increase in the between-class variance $x^TS_b​x$, which corresponds to stronger feature discriminability and better classification performance.
>
> References:
>
> Rao, C.R. (1964). The use and interpretation of principal component analysis in applied research. Sankhya A, 26, 329–358.
>
> Fukunaga, Keinosuke. Introduction to statistical pattern recognition. Elsevier, 2013.
>
> I. T. Jolliffe.  Principal Component Analysis. Springer.  2002.
>
> **Comment 10:**   But the connection of theory on MAD for empirical work on k-NN classifiers still seems odd to me. JL bounds which preserve all distance up to (1+eps) seems a fit for k-NN bounds, not the bounds you provide.
>
> **Response 10:**   As explained in Response 9, maximizing MAD helps enhance the discriminability of projected features, especially when the dataset exhibits sufficiently high discriminability.
>
> As noted in Response 8, the **JL lemma, which focuses on pairwise distance preservation, is *not* ideally suited for analyzing classification performance.** This is because classification accuracy depends mainly on feature discriminability rather than strict distance preservation. **By using MAD analysis instead of JL-based analysis, we can reasonably explain why extremely sparse matrices (with worse JL bounds and thus weaker distance preservation) can still achieve comparable or even better classification performance than dense matrices (with better JL bounds),** as shown in our experiments.

---

### Review · Reviewer_kxkd · 2026-03-11

**Summary Of Contributions:**

1)This work focuses on random projection for sparse {0,±µ}-ternary data using sparse {0,±1} matrices, introducing the mean absolute deviation (MAD) metric to quantify the dispersion of projected features.
2)Theoretical analysis confirms that extremely sparse {0,±1} matrices (e.g., with only one or a few tens of nonzeros per row) can still yield large MAD values, effectively preserving intrinsic data variations.
3)Empirical results demonstrate that these highly sparse structures achieve competitive classification performance while enabling substantial computational savings, thereby validating their efficiency for practical applications.

**Audience:**

Yes

**Audience Explanation:**

This study reveals that the extremely sparse ternary matrices-based random projection can be effectively utilized for dimensionality reduction of ternary quantized data. As random projection is widely applied in machine learning, the results are of interest to researchers working on low-bit quantization, efficient models, and large-scale learning systems.

**Claims And Evidence:**

Yes

**Claims Explanation:**

The claims in the submission are well supported by theoretical analysis and experimental results.

**Requested Changes:**

My main concern is: the paper provides an estimate of the optimal sparsity level of the projection matrix, but it remains unclear whether this estimate can be explicitly computed in practice. This point is particularly important for realworld applications.

---

> ### Author Response · Authors · 2026-03-12
> **Response to Reviewer kxkd**
>
> Dear Reviewer kxkd,
>
> Thank you very much for taking the time to review our work. The question is answered below.
>
> **Comment:**  My main concern is: the paper provides an estimate of the optimal sparsity level of the projection matrix, but it remains unclear whether this estimate can be explicitly computed in practice. This point is particularly important for real-world applications.
>
> **Response:** One  advantage of our theoretical results is that the lower bound of the matrix sparsity $k$ estimated in Theorems 1 and 2 can be explicitly computed, when  both the distribution parameters of the input data and the  convergence error rate $\eta$ are  given.  For instance, by Eq. (9), the lower bound of $k$ is derived as 5078, 1016 and 565 for $q\in\\{0.05,0.25,0.45\\}$ (or equivalently $p\in\\{0.9, 0.5, 0.1\\}$), provided $\mu=1$ and $\eta=0.1$. This shows that  a smaller bound is obtained with a larger $q$ value, which corresponds to a denser data distribution. Note that the above theoretical  bound on $k$  is rather loose; the actual  value is much lower, typically on the order of tens, as verified by the numerical simulation results in Figure 6(a), Appendix B.1. We will incorporate the above discussions into the revised manuscript.

---

### Comment · Reviewer_kxkd · 2026-04-10
**comment**

All issues have been addressed. No more comments

---

### Author Response · Authors · 2026-04-10
**Further clarification about the connection between MAD and classification performance**

Dear Editors and reviewers,

According to the review comments, the main concern is the relationship between MAD (L1-PCA) and classification performance. Specifically, why a larger variance (dispersion) of projected features tends to yield stronger discriminability and better classification accuracy, and whether there exists *mathematical proof*.

Now, we have  identified solid theoretical support for this relationship. In the early studies of  PCA  (Rao, 1964; Fukunaga, 2013: Equation 10.4),    it is established that the total scatter matrix $S_t$​ can be decomposed into the between-class scatter matrix $S_b$​ and the within-class scatter matrix $S_w​$, i.e., $S_t​=S_b​+S_w$​. Based on this fundamental decomposition, as noted in Jolliffe (2002, page 202), it is straightforward to conclude that  when the between-class variance $x^TS_b​x$  (of the projected points) is sufficiently larger than the within-class variance $x^TS_w​x$ (of the projected points), namely  the data exhibit sufficiently high discriminability, the total variance $x^TS_t​x$ (of the projected points) will be dominated by the between-class variance $x^TS_b​x$. In this case,  increasing the total variance $x^TS_t​x$ will directly lead to an increase in the between-class variance $x^TS_b​x$, which corresponds to stronger feature discriminability and better classification performance.

References:

Rao, C.R. (1964). The use and interpretation of principal component analysis in applied research. Sankhya A, 26, 329–358.

Fukunaga, Keinosuke. Introduction to statistical pattern recognition. Elsevier, 2013.

I. T. Jolliffe.  Principal Component Analysis. Springer.  2002.

---

### Decision · Action_Editor_yFPS · 2026-04-12

**Recommendation:** Reject

**Additional Comments:**

This submission received mixed reviews. While one reviewer was positive, the main concern raised by the other reviewers was not fully resolved in the discussion. The central issue is that the paper does not yet sufficiently clarify or justify the connection between its MAD analysis and the classification performance used to motivate the work.

For this reason, I am not able to recommend acceptance in the current round. At the same time, I believe the paper has the potential to become stronger after substantial revision, and I would be open to a future major revision.

A revised version would benefit from either providing a clearer and stronger justification for why the MAD analysis supports the downstream classification claims, or narrowing the presentation so that the contribution is framed more directly as a theoretical study, with the experiments positioned accordingly. It would also help to further clarify the modeling assumptions and improve the overall positioning of the paper.

**Audience:**

Yes

**Audience Explanation:**

This paper studies sparse random projection for ternary/quantized data, a topic that could be of interest to members of the TMLR audience working on efficient machine learning, representation compression, and dimensionality reduction. The analysis of MAD under sparse projection also offers a potentially useful perspective on understanding the behavior of highly sparse random matrices in learning-related settings.

**Claims And Evidence:**

No

**Claims Explanation:**

The paper contains a technically interesting analysis of MAD for sparse random projections. However, based on the reviews, rebuttal, and subsequent discussion, concerns remained about whether the current version provides sufficiently convincing support for the broader learning-related claims emphasized in the paper. In particular, reviewers found the connection between the MAD-based theory and the downstream classification results to be somewhat indirect in the present form.

**Resubmission Of Major Revision:**

The authors may consider submitting a major revision at a later time.